# Mrp1 is involved in lipid presentation and iNKT cell activation by *Streptococcus pneumoniae*

Shilpi Chandra[1], James Gray[2], William B. Kiosses[1], Archana Khurana[1], Kaori Hitomi[1], Catherine M. Crosby[1], Ashu Chawla[3], Zheng Fu [3], Meng Zhao[1], Natacha Veerapen[4], Stewart K. Richardson[5], Steven A. Porcelli[6], Gurdyal Besra [4], Amy R. Howell[5], Sonia Sharma[2,7], Bjoern Peters[8,9] & Mitchell Kronenberg [1,10]

Invariant natural killer T cells (iNKT cells) are activated by lipid antigens presented by CD1d, but the pathway leading to lipid antigen presentation remains incompletely characterized. Here we show a whole-genome siRNA screen to elucidate the CD1d presentation pathway. A majority of gene knockdowns that diminish antigen presentation reduced formation of glycolipid-CD1d complexes on the cell surface, including members of the HOPS and ESCRT complexes, genes affecting cytoskeletal rearrangement, and ABC family transporters. We validated the role in vivo for the multidrug resistance protein 1 (Mrp1) in CD1d antigen presentation. Mrp1 deficiency reduces surface clustering of CD1d, which decreased iNKT cell activation. Infected Mrp1 knockout mice show decreased iNKT cell responses to antigens from *Streptococcus pneumoniae* and were associated with increased mortality. Our results highlight the unique cellular events involved in lipid antigen presentation and show how modification of this pathway can lead to lethal infection.

[1] Division of Developmental Immunology, La Jolla Institute for Allergy and Immunology, La Jolla, CA 92037, USA. [2] The Functional Genomics Center, La Jolla Institute for Allergy and Immunology, La Jolla, CA 92037, USA. [3] Bioinformatics Core, La Jolla Institute for Allergy and Immunology, La Jolla, CA 92037, USA. [4] School of Biosciences, University of Birmingham, Edgbaston, Birmingham B15 2TT, UK. [5] Department of Chemistry, University of Connecticut, 55N. Eagleville Rd, Storrs, CT 06269, USA. [6] Department of Microbiology and Immunology, and Department of Medicine, Albert Einstein College of Medicine, 1300 Morris Park Avenue, Bronx, NY 10461, USA. [7] Division of Cellular Biology, La Jolla Institute for Allergy and Immunology, La Jolla, CA 92037, USA. [8] Division of Vaccine Discovery, La Jolla Institute for Allergy and Immunology, La Jolla, CA 92037, USA. [9] Division of Rheumatology, Allergy & Immunology, University of California, San Diego, La Jolla, CA 92092, USA. [10] Division of Biological Sciences, University of California, San Diego, La Jolla, CA 92093, USA. Correspondence and requests for materials should be addressed to M.K. (email: mitch@lji.org)

Cluster of differentiation 1 (CD1) molecules are non-polymorphic major histocompatibility complex (MHC) class I-like proteins. They are found in most vertebrates and their hydrophobic antigen-binding grooves present lipids rather than peptide antigens[1]. In humans there are four CD1 isotypes: CD1A, CD1B, CD1C, and CD1D, but there is only a single CD1D ortholog in mice[2]. These proteins are expressed as heterodimers consisting of CD1 heavy chains noncovalently paired with β2-microglobulin[3]. CD1 molecules traffick through endosomes, and their distribution in early versus late endosomes differs according to the CD1 isotype[4]. Overall, their localization has more in common with MHC class II than MHC class I intracellular trafficking[5].

Invariant natural killer T cells (iNKT cells) recognize antigens presented by CD1d, and their specificity for bacterial and self-glycolipid antigens is highly conserved[6]. iNKT cells are characterized by the expression of a semi-invariant T cell receptor (TCR) composed of a conserved α chain and a limited repertoire of β chains[7]. These lymphocytes share features with innate immune cells, and they have been widely studied because they influence many types of immune responses in mice and humans[8,9].

While there is much information on the generation and loading of peptides into MHC class I and class II molecules, lipid antigen presentation has been examined less extensively. A few relevant molecules involved in either lipid antigen uptake, carbohydrate processing[10], CD1d intracellular traffic[11], or antigen loading in lysosomal compartments[12–16] have been identified but many relevant steps remain unknown. Mouse CD1d is an excellent prototype for studying CD1-mediated antigen presentation, not only because it stimulates the well-studied iNKT cells but also, as the only mouse CD1 isotype, it recirculates through various compartments, including early and late endosomes and lysosomes. Mouse CD1d first appears on the cell surface by taking a default pathway from the endoplasmic reticulum (ER) to the Golgi apparatus and then to the cell surface. It then is internalized through a process that involves the clathrin-dependent adaptor protein AP-2, and after multiple rounds of recycling, goes to late endosomes and lysosomes in a process mediated by the adaptor AP-3, before finally returning to the cell surface[17–19]. Endosomal trafficking of CD1d is mediated by a YQDI motif in its short cytoplasmic tail, which allows it to interact with the adaptor protein complexes. CD1d can be expressed on the cell surface without this critical motif[19,20], but in that case its presentation of some glycolipids is impaired[10].

In order to obtain a more comprehensive understanding of the pathway leading to lipid antigen presentation by CD1d, we performed a genome-wide small interfering RNA (siRNA) screen in a mouse macrophage cell line loaded with a glycolipid antigen that requires lysosomal carbohydrate removal for its presentation[10]. In this way, we set out to identify genes related to how glycolipid antigens are taken up by antigen-presenting cells (APCs), processed, and loaded into CD1d. Similarly, we wished to characterize genes important for CD1d traffic and surface expression. As a result of the screen, here we identify genes involved in lipid antigen presentation to iNKT cells. These genes are related to vesicular traffic and fusion, and they affect localization of CD1d and/or antigen. Here we show that Abcc1, an ATP transporter, affects CD1d clustering and localization to lipid membrane rafts and is involved in lipid presentation and the protective antibacterial response of iNKT cells.

## Results

**Global screen for lipid antigen presentation**. We performed a genome-wide siRNA screen using a mouse, whole-genome siRNA library to target a total of 17,660 genes, with a pool of four siRNA oligonucleotides per target gene. Because of the large number of cells required, we used transformed APCs and immortalized iNKT cell (hybridoma) responders. The APCs used were a mouse Cd1d1 transfectant of the J774 macrophage cell line (J774-CD1d), which exhibits a high RNAi uptake efficiency and target gene knockdown (Supplementary Fig. 1). The antigen used was galactose (α1-2) α-galctosyl ceramide (GalGalCer), which requires internalization to late endosomes or lysosomes, as well as carbohydrate processing of the disaccharide to a monosaccharide-glycolipid by α-galactosidase A, in order to be recognized[10]. Furthermore, recognition of GalGalCer also requires that CD1d localize to late endosomes/lysosomes, where antigen loading takes place[10]. The use of this antigen therefore allowed us to assess effects on various processes leading to iNKT cell stimulation, including antigen uptake, traffic in APCs to lysosomes, carbohydrate processing, CD1d trafficking to lysosomes, antigen loading, and movement of CD1d-lipid antigen complexes to the cell surface. After siRNA knockdown, J774-CD1d cells were exposed to GalGalCer, and the ability of an iNKT cell hybridoma to produce interleukin (IL)-2 in response to antigenic stimulation provided a convenient and robust readout for antigen presentation (Fig. 1a).

In the primary whole-genome screen, by excluding siRNA knockdowns that significantly diminished APC viability, we identified 1027 genes in the "down" category (Fig. 1b), corresponding to <30% of the control IL-2 release remaining following gene depletion (Supplementary Data 1). We focused on those genes that caused decreased CD1d-mediated antigen presentation after knockdown and carried out a secondary re-screen of 1027 candidate genes. In general, due to assay variability, the degree of IL-2 suppression in the secondary screen was lower. By using 50% knockdown as a cutoff, we thereby validated 424 hits (Supplementary Data 2).

**Functional classification**. To functionally annotate the identified genes, we classified them based on Gene Ontology (GO) analysis, employing Cytoscape[21]. Based on this analysis, 70 categories of cellular processes and structures were identified as participating in a positive way in CD1d-mediated antigen presentation (Supplementary Data 3). The processes identified include 14 GO classes likely to be directly related to the cell biology of antigen presentation, such as those including transport, lysosomes, endosomes, and organelle organization. These categories are listed in Supplementary Table 1 along with the list of genes in each category. A total of 207 genes (49%) were identified in these GO categories, and prioritized for further study. Classes that are related less directly to the cell biology of antigen presentation, but that could have important effects, including transcription, metabolism, and signal transduction, were also identified.

To reduce false positives, we verified expression of the hits in primary macrophages; using a microarray analysis comparing expression of genes in J774-CD1d cells to peritoneal macrophages from BALB/c mice, the strain of origin of the J774 cells. The expression patterns were highly similar and of the candidate genes identified, out of 207 genes, 141 were found to be expressed above the minimum cutoff, as described in Methods, by both cell types.

To analyze for off-target effects, we performed a deconvolution screen by testing each of the four single siRNAs from each oligonucleotide pool. We used the criteria of >50% inhibition of antigen presentation by at least two of the four siRNAs for further inclusion. As a result, we identified 48 genes as having the most robust influence (Fig. 2) listed in Supplementary Data 4, including several previously reported to be important for

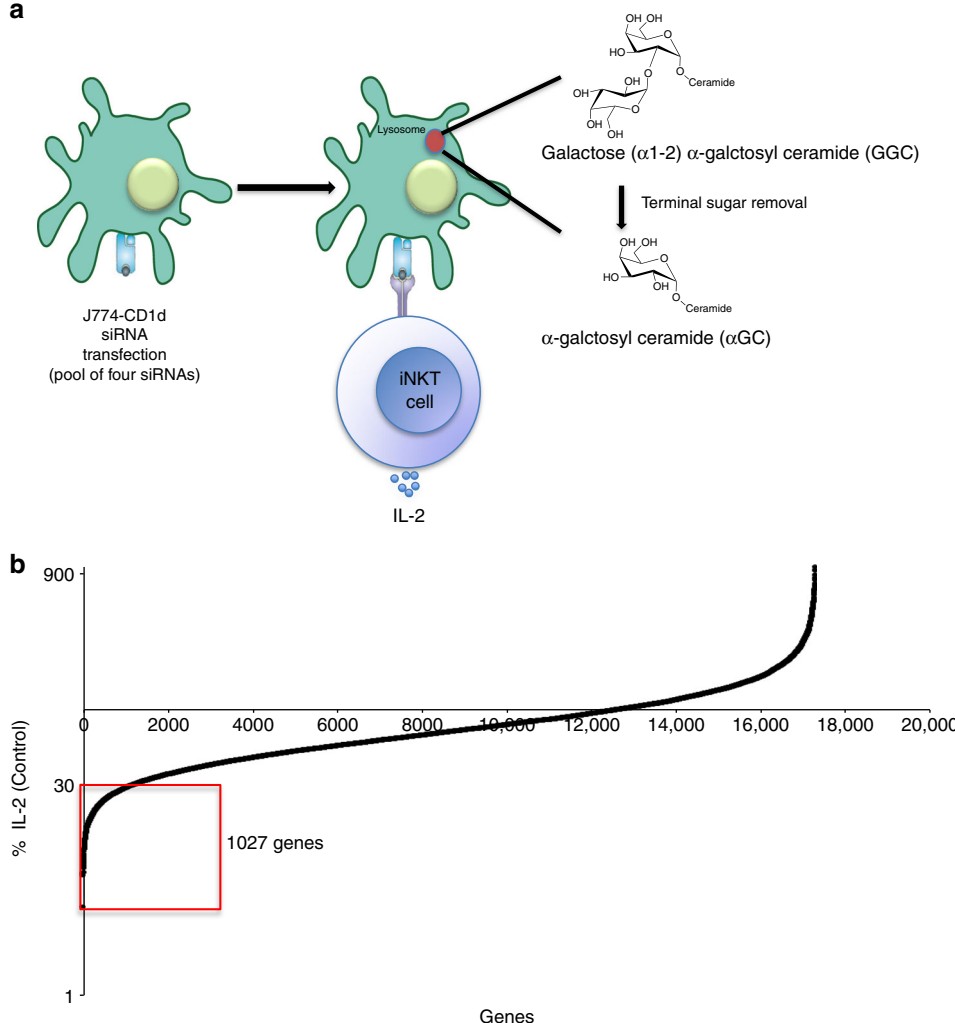

**Fig. 1** Data analysis of the siRNA screen. **a** Schematic outline of the assay. J774-CD1d cells were transfected with siRNA pools, 36 h later these cells were loaded with GalGalCer for 6 h before culture with an iNKT cell hybridoma. Sixteen hours later supernatant was used to perform IL-2 Alpahlisa. **b** Plots IL-2 release stimulated by APCs transfected with each of the 17,660 siRNAs in the library ordered by percentage of control (Log). Genes showing >70% decrease in IL-2 (average of two wells) as compared to the control (red box) were considered for further analysis. See also Supplementary Fig. 1

presentation of lipid antigens by CD1d, including *B2m*, *Psap*, encoding the lysosomal protein prosaposin, which is important for glycosphingolipid (GSL) metabolism, and *Vps11*, a member of the homotypic fusion and protein sorting (HOPS) complex important for tethering vesicles for fusion[12,22–26].

To determine the likely protein–protein interactions among the identified genes, we used knowledge-based Ingenuity Pathway Analysis (IPA; Ingenuity Systems). Three functional networks were identified containing >5 genes that included 46 out of 48 identified genes and their interacting partners. A network involved in protein and vesicle trafficking encompassed 23 identified genes (Fig. 3), including genes in the HOPS and soluble *N*-ethyl maleimide sensitive factor attachment protein receptor (SNARE) complexes involved in vesicle tethering and fusion, as well as other genes such as *Rhob* and *Snx20*, also involved in protein trafficking. The other networks identified (Supplementary Fig. 2) include lipid antigen presentation (12 identified genes) and cell morphology and interactions (11 identified genes). Lipid antigen presentation includes *Cd1d, B2m, and Psap*, already known to be involved in this process and genes involved in lipid metabolism (*Abcc1, Cln8, Gal3st, Glb1, and Psap*). The cell morphology and cell–cell signaling and interaction network includes *Dock2, Elmo1, Prad6a, and Was*.

A number of the selected genes have been shown to be related to disease states (using diseases tool—diseases.jensenlab.org) such as lysosomal storage diseases, including Niemann-Pick (*Glb1* and *Gal3st*), Fabry (*Psap*), Krabbe disease (*Glb1* and *Psap*), Hermansky-Pudlak syndrome (*Tsg101* and *Vps16*), some of which have been associated with deficient iNKT cell differentiation as well as defective antigen presentation[27]. Several of the genes identified are reported to be involved in infections, autoimmune and inflammatory diseases, or cancer (Supplementary Table 2). We were not able to identify all the genes previously reported to be involved in CD1d-mediated antigen presentation, however, perhaps because the knockdowns were not complete[2,11,13,25].

**Predominant effects on CD1d-lipid antigen complexes**. Reducing the supply of peptide antigens markedly decreases surface expression of MHC class I proteins[28], but we found few genes that decreased surface expression of CD1d. Of the 48 selected genes, significantly decreased CD1d expression was observed only in a few cases. These included the gene encoding cytoskeletal protein Pfn3; Cyp4b1, which is an endoplasmic reticulum protein and is known to be involved in metabolism and synthesis of

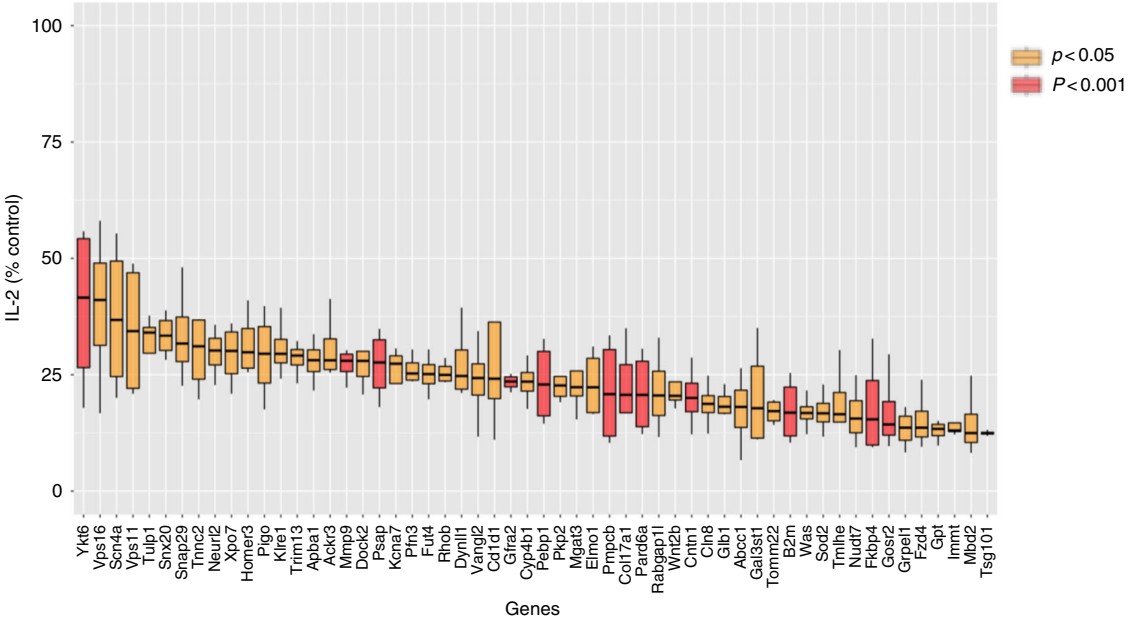

**Fig. 2** Decreased lipid antigen presentation after knockdown of identified genes. J774-CD1d cells were transfected with siRNA pools, 36 h later these cells were loaded with GalGalCer for 6 h before culture with an iNKT cell hybridoma. Sixteen hours later supernatant was used to perform IL-2 Alpahlisa. Plot of normalized IL-2 release for selected hits compared to control. An average of three experiments is shown. Boxes represent the interquartile range; maximum and minimum values are displayed with vertical lines. *p*-Values are indicated (paired *t*-test)

sterols; and Fdz4, a member of frizzled gene family (Fig. 4a). We analyzed selected genes affecting CD1d presentation for their influence on surface expression of MHC antigen-presenting molecules. Knockdown of several of the selected genes that reduced CD1d surface expression, such as *Pfn3*, also caused a statistically significant decrease in surface expression of K$^d$ and D$^d$ MHC class I molecules (Supplementary Fig. 3A). Similarly, regarding effects on MHC class II surface expression, only knockdown of *Fut4*, *Psap*, and *Pigo* caused decreased MHC-II expression (Supplementary Fig. 3B).

CD1d loads phospholipid antigens in the ER and acquires GSL antigens predominantly in lysosomes. To determine if complexes of CD1d with αGalCer were diminished following gene knockdown, despite equivalent amounts of surface CD1d expression, we made use of the monoclonal antibody (mAb) L363 that specifically detects αGalCer-CD1d complexes[29] (Supplementary Fig. 4). Knockdown of most of the selected 48 genes reduced the amounts of antigen-CD1d complexes on the cell surface detectable by flow cytometry (Fig. 4b). The molecules that fall in this category include several cytoskeletal proteins, proteins involved in vesicular and cargo traffic in endosomes and lysosomes, and proteins present in mitochondria or the plasma membrane. For 10 of the genes, however, the decrease in staining for αGalCer-CD1d complexes was not statistically significant, perhaps reflecting variability in the knockdown and mAb staining in different experiments. It is also possible that some of these knockdowns affected the concentration of CD1d into lipid rafts or nanoclusters on the plasma membrane, which is important for iNKT cell stimulation[30], without diminishing the total lipid antigen-CD1d detectable with L363 on the cell surface.

CD1d and MHC-II antigen presentation pathways both depend on antigen loading in acidic late endosomal or lysosomal compartments, therefore we analyzed the role of the the 48 identified genes in MHC class II antigen presentation to assess the extent to which the two pathways overlap. After siRNA-mediated knockdown, J774-CD1d cells were incubated with ovalbumin (OVA) and cultured with CD4$^+$ T cells isolated from OVA-specific DO11.10 × *Rag*$^{-/-}$ TCR transgenic mice. Our results

indicate that only some of the gene knockdowns affected MHC class II antigen presentation to the extent that CD1d antigen presentation was affected (<30%), and only 9 reached <50% when compared to the control (Supplementary Fig. 5). Among the genes most affecting MHC class II presentation by the J774-CD1d cells were the *Rhob*, *Mbd2*, and *Tsg101*. These three genes have been shown to affect MHC-II antigen presentation previously[31–33]. Therefore, the pathway leading to GSL antigen uptake, processing, and presentation by CD1d is in part distinct from protein processing and presentation by MHC class II.

**Effects on lipid antigen uptake**. To better understand the specific role(s) of the molecules identified in the screen, we analyzed the effect on antigen localization, by using confocal microscopy to follow the uptake of Bodipy-labeled αGalCer into lysosomes of J774-CD1d cells. We analyzed four genes that affected antigen-CD1d complex formation without decreasing CD1d surface expression significantly (Fig. 5a and Supplementary Fig. 6), some of which were not known previously to be involved in antigen presentation. These included the following: (1) Dedicator of cytokinesis 2 (Dock2), a guanine nucleotide exchange factor involved in actin polymerization that influences several processes, including phagocytosis and antigen uptake by dendritic cells (DC)[34]. (2) Synaptosome-associated protein 29 (Snap29), which is part of the SNARE complexes involved in membrane fusion. It has been reported to be involved in mast cell phagocytosis[35], but has not been associated with antigen presentation. (3) Tumor susceptibility gene 101 (Tsg101), which is a component of the endosomal sorting complexes required for transport I (ESCRT-I) that binds to ubiquitinated cargo proteins and is required for their sorting into multivesicular bodies and exocytosis[36]. It may be involved in the trafficking of MHC class I molecules that are ubiquitinated and degraded following Kaposi sarcoma virus infection[37], and also plays a role in mannose receptor-mediated cross-presentation of antigens in the class I pathway[38]. (4) Vacuolar protein sorting-associated protein homolog 11 (Vps11) a HOPS complex member involved in endosome-lysosome fusion, which was previously reported to be

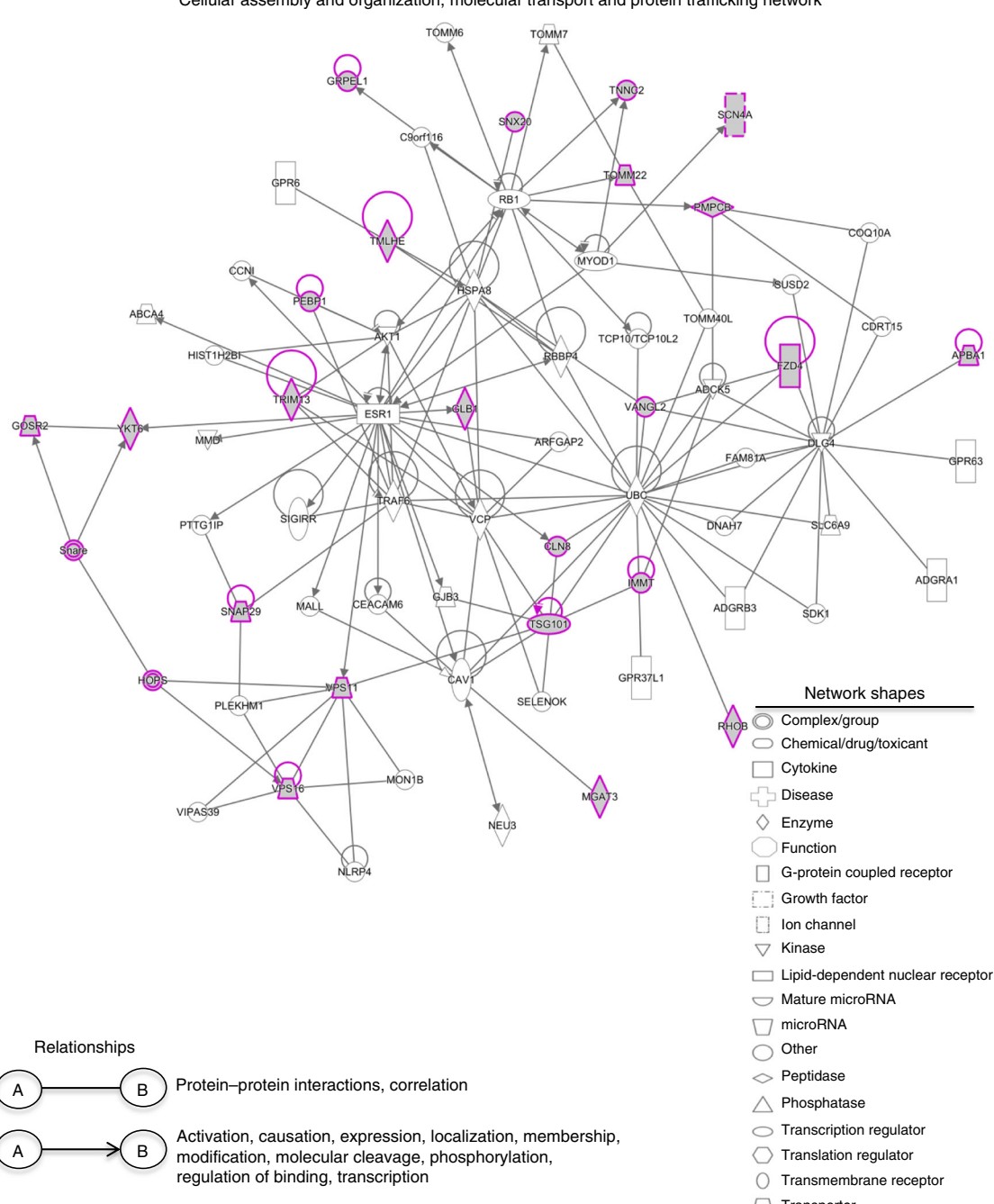

**Fig. 3** Network analysis of the identified genes. Three major networks containing >5 genes were identified using IPA. Figure represents one out of three networks that include genes with functions related to cellular assembly and organization, molecular transport, and protein trafficking. See also Supplementary Fig. 2

important for lipid antigen presentation by mouse CD1d[25]. A significant knockdown was observed at the protein level for all the four genes (Supplementary Fig. 7). In the absence of specific gene knockdown, we observed that the maximum accumulation of labeled antigen in lysosomes was reached at 2 h. Knockdown of *Snap29* or *Tsg101* greatly reduced localization of antigen to lysosomes, knockdown of *Vps11* delayed antigen localization, but knockdown of *Dock2* had no effect (Fig. 5b, c).

**Effects on CD1d localization**. Because trafficking of CD1d to lysosomes is important for the acquisition of GSL antigens[2,13,27],

we used confocal microscopy to obtain a quantitative analysis of the localization of CD1d in the J774-CD1d cells at steady state. We used Rab5 as a marker for early endosomes and Lamp1 for late endosomes and lysosomes. Knockdown (Supplementary Fig. 7) of *Snap29*, *Vps11* and *Dock2* led to a significantly decreased localization of CD1d into Lamp1+ compartments (Fig. 6a, c), and for Vps11 and Dock2, there was a significant increase in the accumulation of CD1d in Rab5+ early endosomes (Figs. 6b and 7). Reduced expression of Tsg101 did not lead to a highly altered intracellular distribution of CD1d, although the size and distribution of Lamp1+ compartments appeared different in some frames, the difference was not significant when many cells

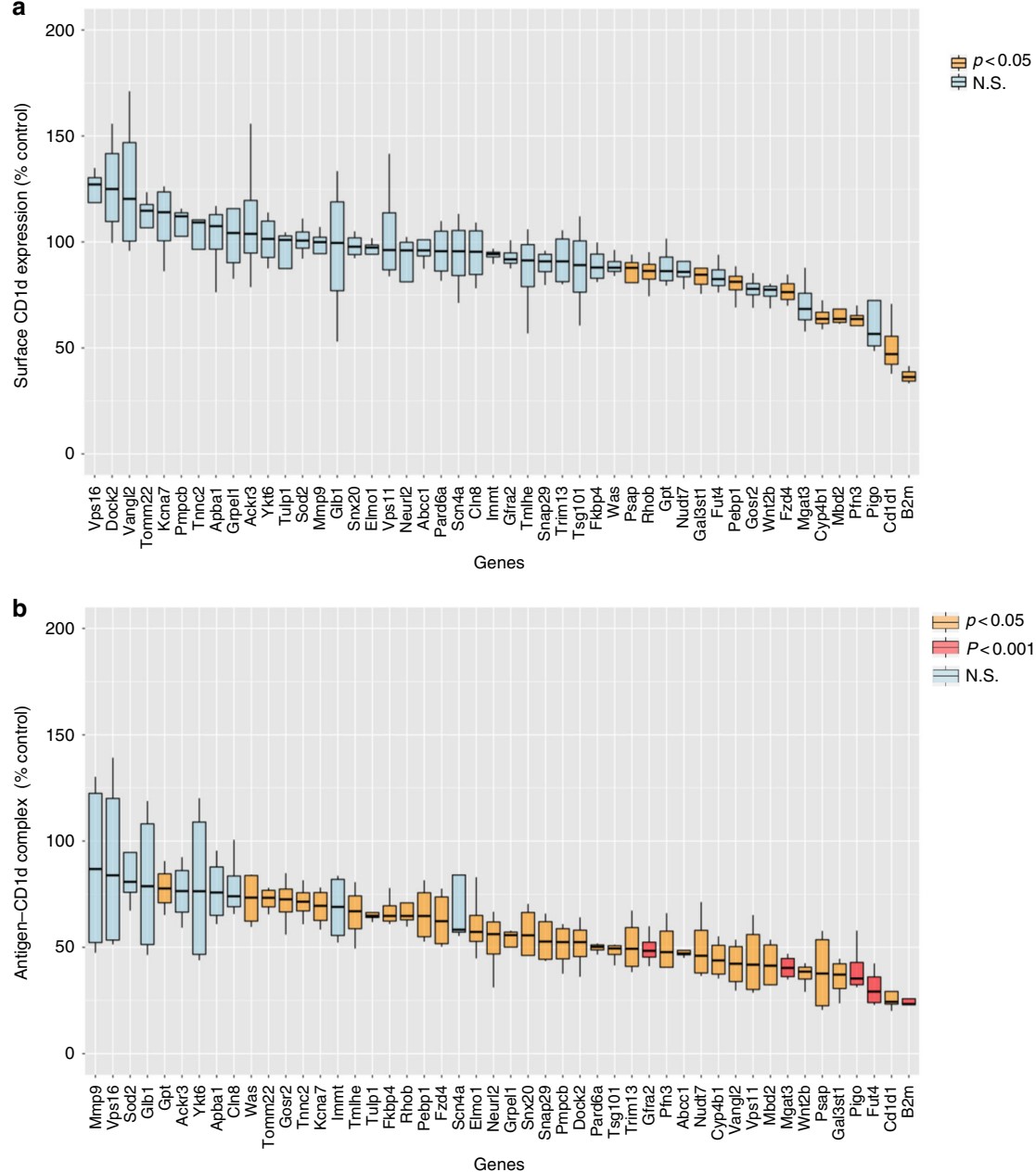

**Fig. 4** Gene knockdown effects on surface CD1d and CD1d-lipid complex formation. **a** Effect on CD1d surface expression assessed by flow cytometry. The figure represents percentage change in mean fluorescence intensity (MFI) as compared to a non-targeting control. Average value from four experiments is shown. **b** Flow cytometry analysis of surface expression of αGalCer-CD1d complexes by J774-CD1d cells (live, singlets) after GalGalCer loading. Normalized MFI compared to non-targeting control, the average of four experiments is shown. Boxes represent the interquartile range; maximum and minimum values are displayed with vertical lines. p-Values are indicated (paired t-test). See also Supplementary Fig. 3, 4, 5 and 6

were quantitated (Supplementary Fig. 8). As summarized in Supplementary Fig. 9, genes that affect the ability of CD1d to localize in lysosomes, such as *Dock2*, *Snap29*, and *Vps11*, or that affect the ability of GSL antigens to traffic there, such as *Snap29* and *Tsg101*, impaired the presentation of GalGalCer without diminishing the surface expression of CD1d.

**Abcc1 deficiency results in diminished iNKT cell activation.** Although αGalCer is internalized to late endosomes in APC[39], and presentation of αGalCer is enhanced when it is taken up by cells[40], this antigen also can load into CD1d on the cell surface, and unlike GalGalCer, can be presented by cells expressing a form of CD1d

that cannot localize to endosomes[39]. Therefore, the genes that affect the response to GalGalCer more than αGalCer, likely function in either carbohydrate processing or antigen and/or CD1d trafficking to lysosomes. The response to GalGalCer compared to the response to αGalCer of the 48 hits was >1.5-fold higher for 13 of the genes, and 2-fold greater for 5 including *Abcc1*, which had the highest increase (Fig. 8a). In every case, however, the knockdowns also inhibited the αGalCer response to some extent (Supplementary Fig. 10). Among these GalGalCer-selective genes were *Elmo1*, previously shown to be involved in phagocytosis[41], and *Psap*, important for lipid loading of CD1d in lysosomes[42–44]. *Abcc1*, which encodes the multidrug resistance protein 1 (Mrp1), intrigued us because Mrp1 is a member of the ATP-binding cassette (ABC)

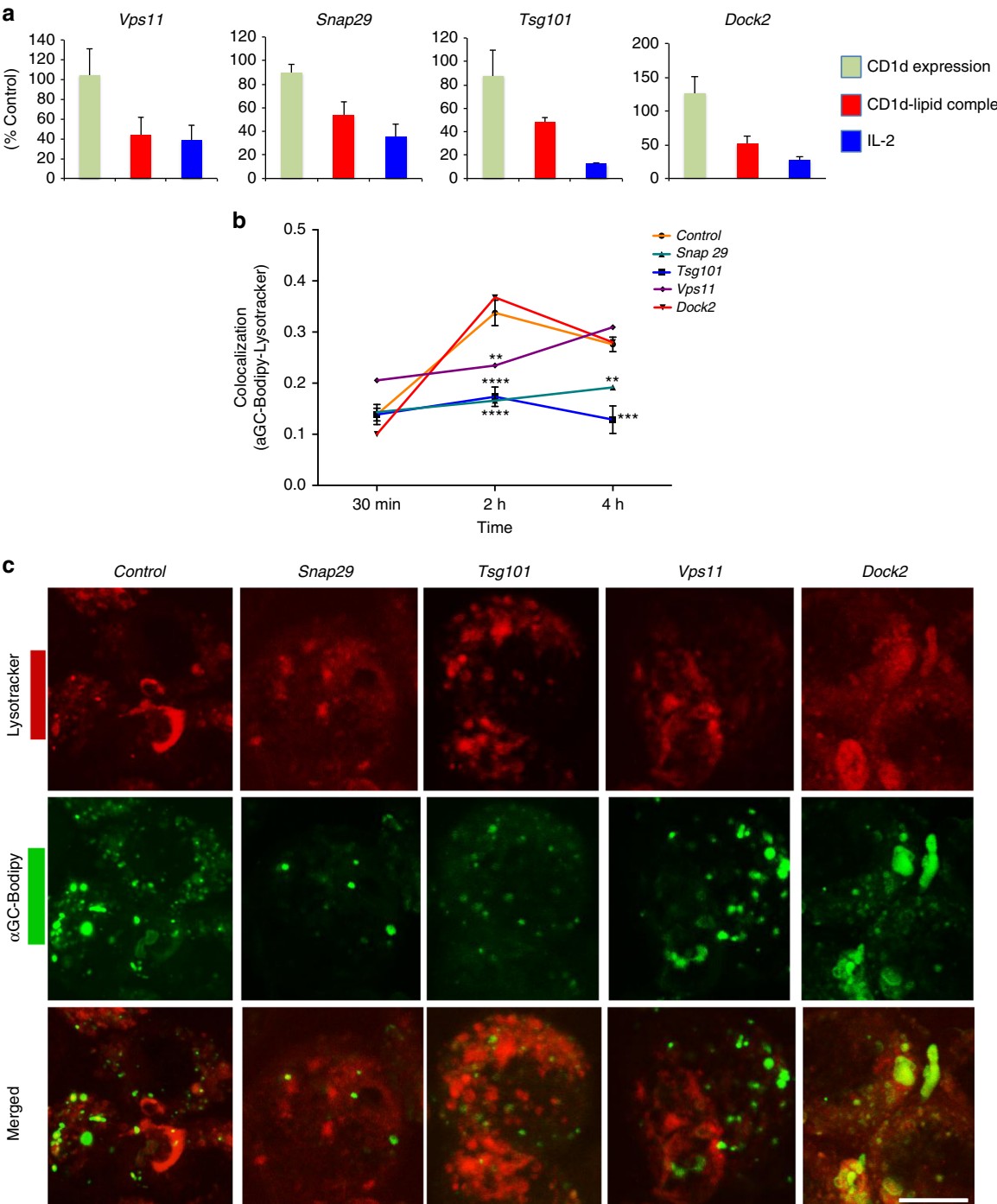

**Fig. 5** Effects on lipid antigen uptake to lysosomes. **a** Comparative plots of normalized values for CD1d expression, CD1d-lipid complex formation by flow cytometry, and IL-2 release as described above for indicated genes. **b** Labeled antigen uptake. APCs with the indicated genes knocked down were incubated with αGalCer-Bodipy for the indicated times and LysoTracker added for a 2 h chase. Figure represents the co-localization coefficient. Representative data from one of two experiments in which at least 100 cells per time-point were analyzed. Nonsignificant *p*-values are not indicated. **c** Representative images for αGalCer-Bodipy and LysoTracker at 2 h time-point. Scale bar 10 μm. Graphs represent mean ± SD. **\*\****p* < 0.01; **\*\*\****p* < 0.001; **\*\*\*\****p* < 0.0001 (one-way ANOVA). See also Supplementary Fig. 7, 8, and 9

family of transporter molecules, which also includes the transporter associated with antigen processing (TAP), required for peptide loading and surface expression of MHC class I molecules.

Like the majority of the other hits, knockdown of *Abcc1* led to reduced antigen-lipid complex without changing CD1d surface expression (Figs. 4 and 8b, c). To validate a role for this protein, we analyzed peritoneal macrophages from *Abcc1*$^{-/-}$ mice for their ability to present lipid antigens to an expanded line of mouse iNKT cells. We observed peritoneal macrophages were defective in antigen presentation of αGalCer and GalGalCer (Fig. 8d). To test antigen presentation function in vivo, we injected αGalCer and measured interferon gamma (IFNγ) and IL-4 in the sera by enzyme-linked immunosorbent assay (ELISA). The response by *Abcc1*$^{-/-}$ mice was lower compared to the FVB parental strain (Fig. 8e). Despite this, the percentage of iNKT cells in thymus and spleen was undiminished in *Abcc1*$^{-/-}$ mice, and

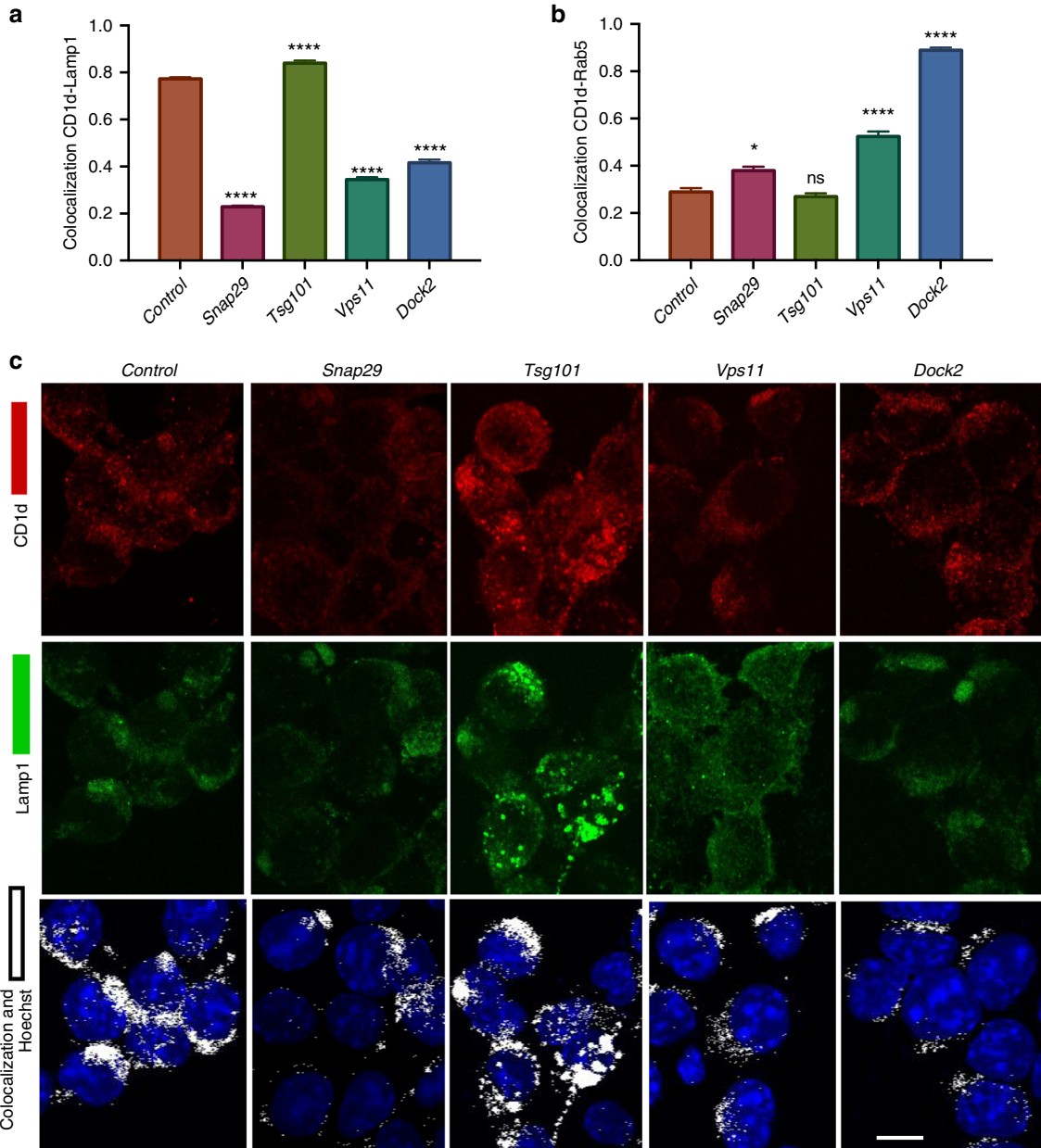

**Fig. 6** Effects on CD1d trafficking to the lysosome. **a** Co-localization coefficient of CD1d with the late endosomal marker, Lamp1 or early endosomal marker Rab5 after knockdown of the indicated genes in J774-CD1d cells. Representative data from one of two experiments in which at least 200 cells per condition were analyzed. **b** Representative images for the CD1d and LAMP1 co-localization after knockdown of the indicated genes in J774-CD1d cells. Scale bar 10 μm. Graphs represent mean ± SEM. *$p < 0.05$; ****$p < 0.0001$ (one-way ANOVA). See also Supplementary Fig. 7, 8, and 9

the percentage in the liver was decreased only moderately (Supplementary Fig. 11).

Because antigen-dependent activation of iNKT cells is critical for protection of mice from *S. pneumoniae*[45,46], we analyzed activation of iNKT cells after infection and the survival of *Abcc1*[−/−] mice. The method for gating of lung iNKT cells is shown in Supplementary Fig. 12. Production of IFNγ by TCR-activated iNKT cells was reported to be protective following *S. pneumoniae* infection[46], and in *Abcc1*[−/−] mice, this response was blunted (Fig. 8f). Furthermore, *Abcc1*[−/−] mice had decreased survival and increased bacteria in their lungs at day 2 after infection (Fig. 8g). Therefore, in vitro and in vivo data support the conclusion that lipid antigen presentation by CD1d is decreased in primary APC from *Abcc1*[−/−] mice, and this likely has an effect on the host protective immune response

early after infection, although effects on cell types other than iNKT cells were not excluded.

**Abcc1 deficiency results in altered CD1d distribution**. To identify the mechanism(s) by which Abcc1 acts on lipid antigen presentation, we analyzed antigen and CD1d trafficking in peritoneal macrophages from gene-deficient mice, as described above. The expression of CD1d on the surface of F4/80[+] peritoneal macrophages from *Abcc1*[−/−] mice was not reduced (Supplementary Fig. 13A). When the accumulation of αGalCer-Bodipy in lysosomes was measured, there was only a moderate decrease, especially at the 2 h time (Supplementary Fig. 13B). We observed a much more striking difference in the distribution and

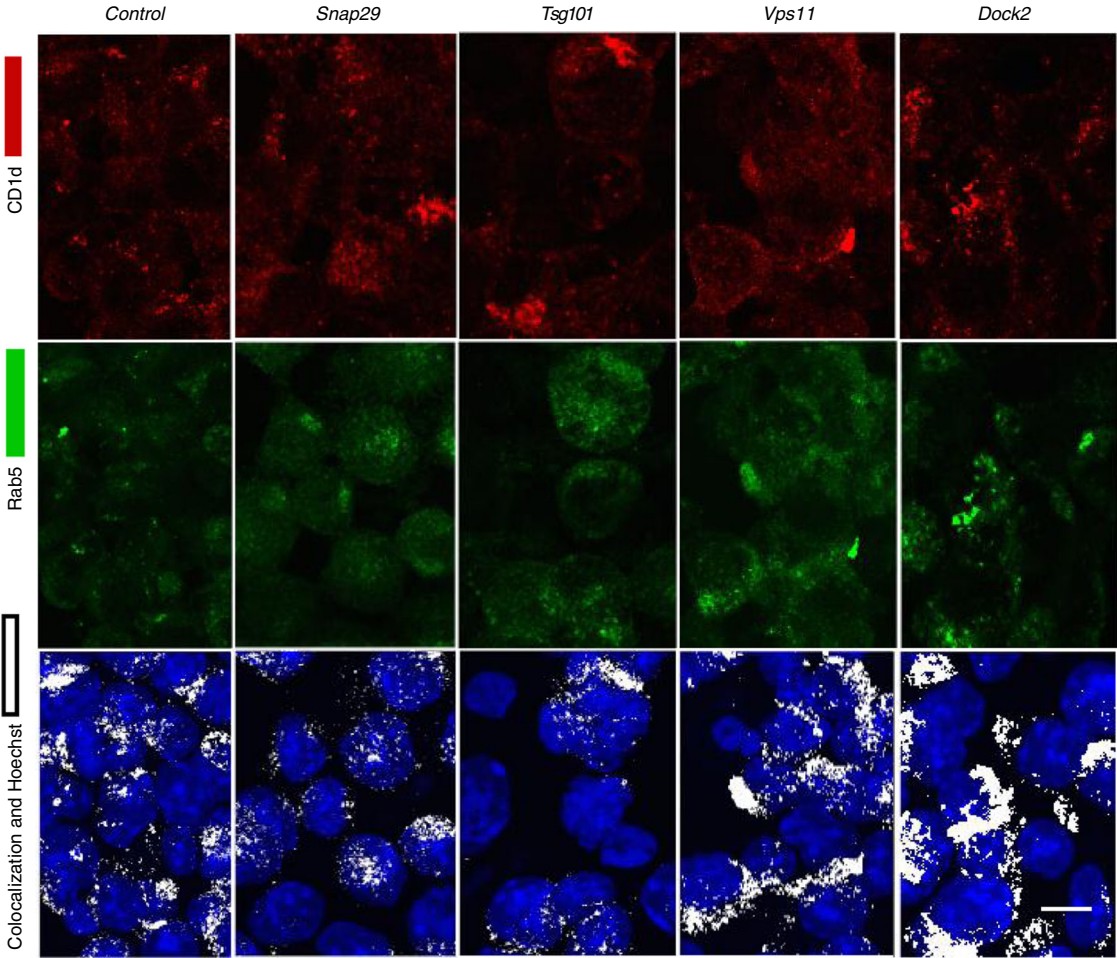

**Fig. 7** Effects on localization of CD1d to early endosomes. Representative images for CD1d and Rab5 co-localization after knockdown of the indicated genes in J774-CD1d cells. Representative data from one of two experiments in which at least 200 cells per condition were analyzed. Scale bar 10 μm. See also Supplementary Fig. 9

localization of CD1d. There was a reduction in CD1d co-localization with Lamp1$^+$ late endosomal compartments (Fig. 9a). Furthermore, the average size of the Lamp1$^+$ vesicles in macrophages from $Abcc1^{-/-}$ mice was decreased (Fig. 9b). These alterations in late endosomes and lysosomes likely contribute to the pronounced effect of $Abcc1$ knockdown on the response to GalGalCer.

Abcc1 deficiency also led to alterations in the plasma membrane that likely affected lipid antigen presentation. In macrophages from $Abcc1^{-/-}$ mice, the density of lipid rafts in the membrane was decreased (Fig. 10a, b). Furthermore, CD1d co-localization with lipid rafts was diminished, and the size of CD1d clusters on the cell surface was reduced (Fig. 10a, c). Therefore, altered membrane biosynthesis in cells from $Abcc1^{-/-}$ mice is likely an underlying cause of changes in CD1d distribution that led to reduced antigen presentation.

## Discussion

As a model system to gain deeper insight into the presentation of nonpeptide antigens, we analyzed the genetic control of the presentation of GSL antigens by CD1d to iNKT cells, a T-cell type that is prominent because of its distinct behavior and influence on many types of immune responses[47]. Few of the selected knockdowns decreased surface CD1d expression, and instead, the majority decreased the formation of αGalCer complexes with CD1d. Long-lived CD1d molecules recycle from the cell surface

through early endosomes, followed by traffic to lysosomes before return to the cell surface[17,48]. Uptake of GSL antigens, their traffic to lysosomes, carbohydrate processing, and loading into CD1d is also a multifaceted process. Therefore, a decrease in the formation of GSL-CD1d complexes on the cell surface could be due to deficit in any one or more of these processes.

Consistent with the requirement for complex intracellular trafficking events, our results implicated components involved in various parts of the endosomal and lysosomal systems, including ESCRT-I, HOPS, and SNARE complexes, and as well as proteins involved in cytoskeletal rearrangements. Tsg101 is a member of the ESCRT-I complex that binds to ubiquitinated cargo proteins. Although human CD1d can be ubiquitinated on a lysine in its cytoplasmic tail[49], mouse CD1d does not have this lysine, and therefore the action of Tsg101 likely is not directly on the CD1d protein. Consistent with this, the primary effect of $Tsg101$ gene knockdown was impaired localization of GSL antigen to lysosomes. Knockdown of HOPS complex member $Vps11$ caused a delay in antigen traffic to lysosomes and also reduced CD1d localization there. Our data are consistent with an earlier report showing similar effects of knockdown of HOPS complex regulator Arl8b[25]. Knockdown of SNARE complex gene $Snap29$ led to a pronounced reduction in antigen trafficking to lysosomes and decreased CD1d in LAMP1$^+$ vesicles. Although ESCRT, HOPS, and SNARE complex proteins all have been reported to regulate autophagy[50], knockdown of macroautophagy genes, such as $Atg7$,

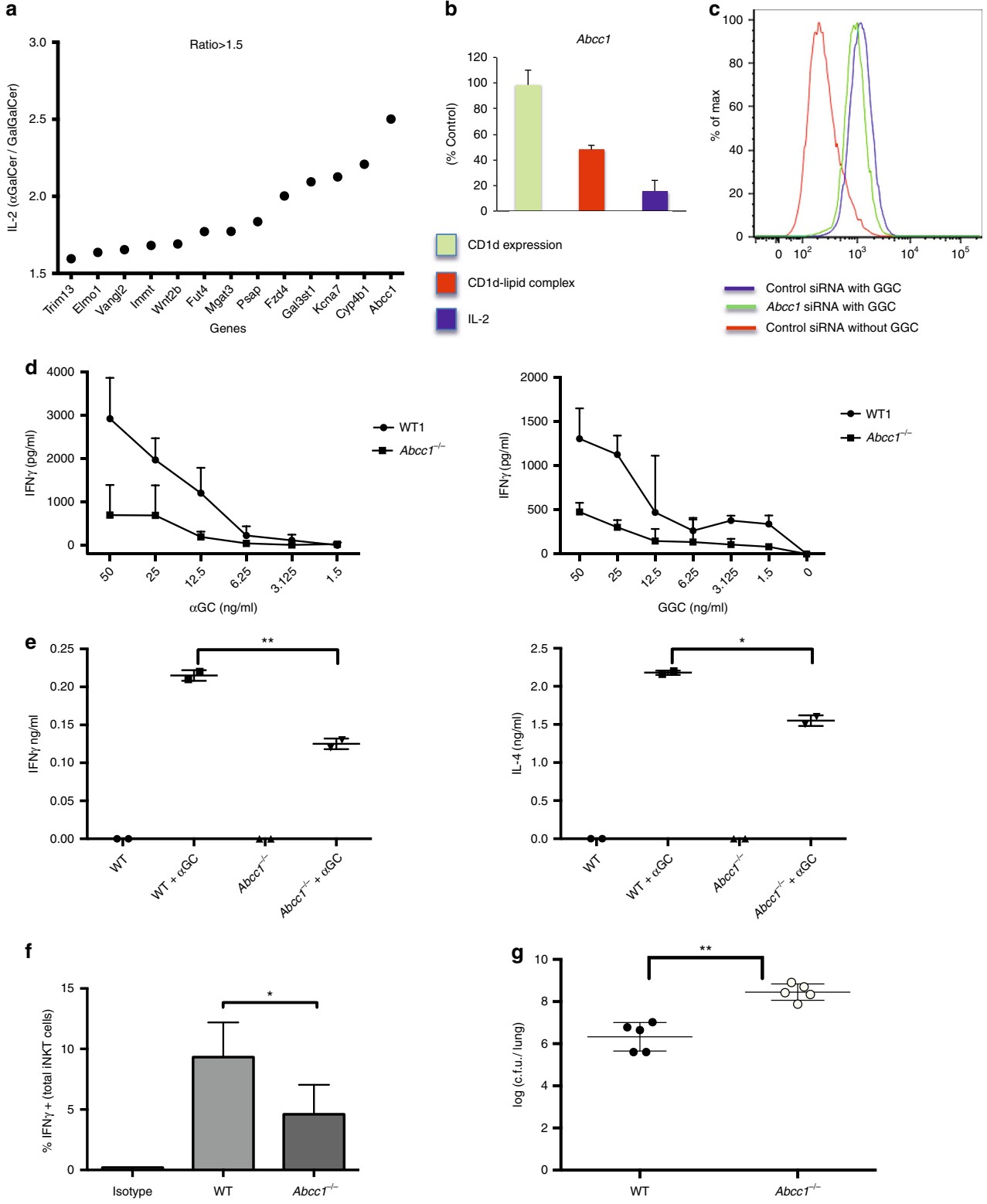

did not have a strong effect on iNKT cell responses, consistent with a recent report[51]. A more recent study, however, indicated a role for Atg5 in inhibiting CD1d antigen presentation[52]. Similar to *Vps11* and *Snap29*, *Dock2* gene knockdown also reduced CD1d accumulation in lysosomes. Dock2 affects the actin cytoskeleton, which recently has been shown to regulate the efficiency of antigen presentation by affecting the size and mobility of CD1d

nanoclusters on the cell surface[30]. Elmo1, involved in phagocytosis[53], interacts constitutively with Dock2 in some cell types[54] and also was identified by the screen.

We found that gene knockdown of several members of the ABC transporter family had effects on lipid antigen presentation, although only knockdown of the *Abcc1* gene, which encodes Mrp1, met our criteria for deeper investigation. We confirmed the

**Fig. 8** Role of Abcc1 in lipid antigen presentation. **a** After knockdown, J774-CD1d were treated with either αGalCer or GalGalCer and cultured with an iNKT cell hybridoma before measuring IL-2. Figure represents the fold change in IL-2 secretion comparing the normalized response to αGalCer with GalGalCer. Genes that showed >1.5-fold more IL-2 in case of αGalCer are plotted. Figure represents the average of two experiments. **b** Comparative plots of surface CD1d expression, CD1d-lipid complex formation, and IL-2 release after knockdown of Abcc1 in J774-CD1d cells. Figure represents the average of three experiments. **c** Figure represents the decrease in surface CD1d-lipid complex expression by J774-CD1d cells after knocking down Abcc1 after loading with GalGalCer. **d** Peritoneal macrophages were isolated from wild-type (WT) and Abcc1−/− mice, loaded with αGalCer and GalGalCer at the indicated concentrations and incubated with a mouse iNKT cell line before IFNγ measurement. Figure shows the representative plots from one of two experiments. **e** WT and Abcc1−/− mice were injected i.v. with αGalCer and IFNγ and IL-4 was measured after 2 h of injection. Figure shows the representative plots from one of two experiments. **f** WT and Abcc1−/− mice were infected with 2.5 × 10^6 S. pneumonia for 14 h followed by intracellular cytokine staining of gated, lung iNKT cells (See Supplementary Fig. 12). Figure represents the percent IFNγ positive cells in uninfected, and infected WT and Abcc1−/− mice. Figure shows the representative plots from one of two experiments in which pooled cells from two mice for FVB and three mice for Abcc1−/− mice were analyzed. **g** WT and Abcc1−/− mice were infected with 0.5 × 10^5 S. pneumonia and after 2 days lung CFU were measured. Data are representative from one of the three experiments. Graphs represent mean ± SD. *p < 0.05; **p < 0.01 (one-way ANOVA). See also Supplementary Fig. 10 and 11

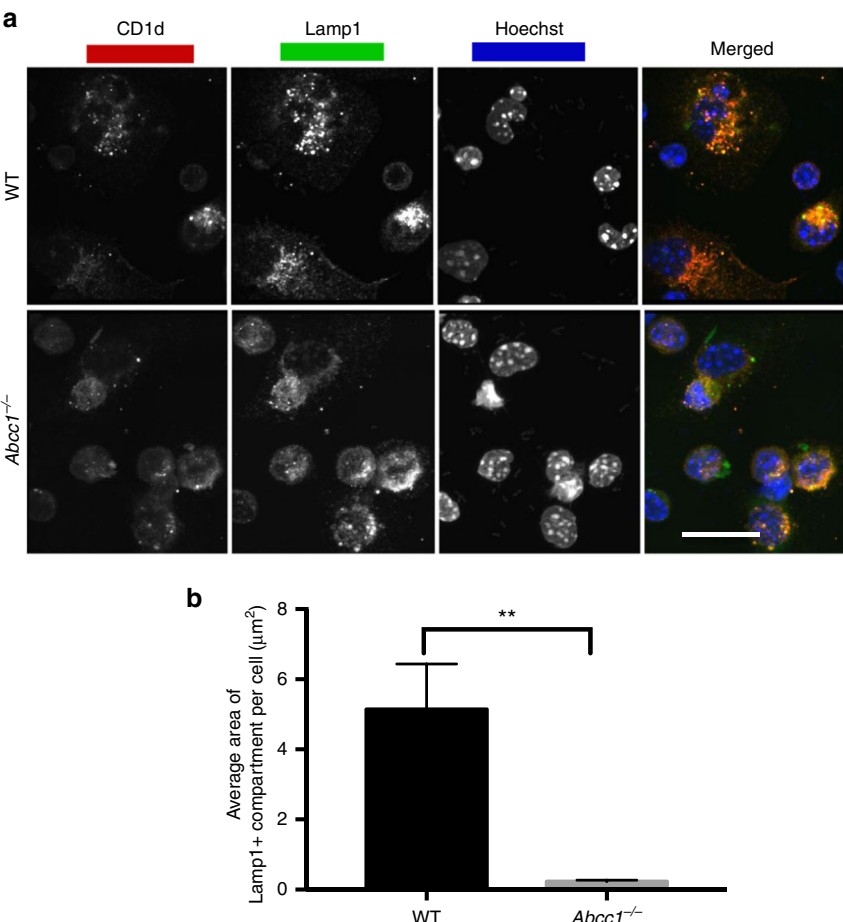

**Fig. 9** Reduced CD1d-Lamp1+ vesicles in macrophages from Abcc1−/− mice. **a** Peritoneal macrophages were stained with CD1d and Lamp1 antibodies along with Hoechst and images were acquired using confocal microscopy. Representative images from one of two experiments with at least 100 cells per condition analyzed. **b** Average area of Lamp1+ compartments in WT and Abcc1−/− mice. Scale bar 20 μm. Graphs represent mean ± SEM. **p < 0.01 (one-way ANOVA). See also Supplementary Fig. 13

role of Mrp1 in CD1d antigen presentation in vitro and in vivo, which correlated with reduced iNKT cell activation and increased susceptibility of Abcc1-deficient mice to S. pneumoniae. In contrast to our study, a previous publication reported that Mrp1-deficient mice were protected against S. pneumonia[55]. This difference might be attributed to the different serotypes of S. pneumonia used, as some strains might be less dependent on iNKT cells for protection. Furthermore, there were differences in the dose and route of infection, retropharyngeal in our experiments as opposed to intranasal. A study on Mycobacterium tuberculosis agreed with the outcome of our work in showing that

Mrp1-deficient mice were more susceptible to infection by inhalation, although the role of iNKT cells to susceptibility was not examined[56]. Also, differences in the background of the mice used must be considered, as the study by Schulz et al. used (129/Ola)/FVB (50%/50%) mice, whereas the mice that we used and in Verbon et al. were pure FVB background. We recently reported that another ABC family transporter, Abca7, also regulated the number of lipid rafts and the co-localization of CD1d in lipid rafts[57]. Unlike Mrp1, however, there was reduced CD1d surface expression in double-positive thymocytes in Abca7−/− mice, providing a connection to the decreased positive selection of

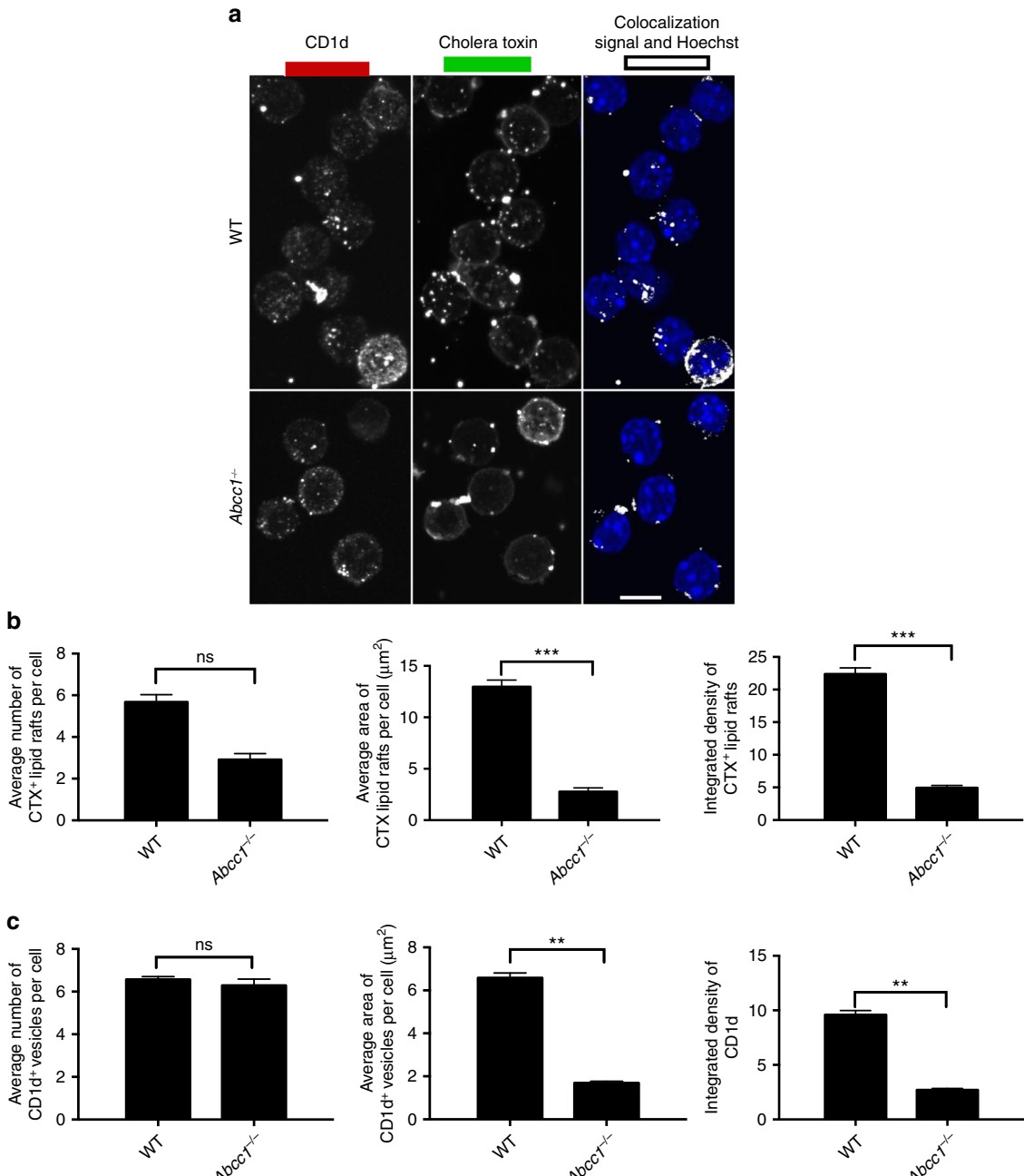

**Fig. 10** Alterations in CD1d and lipid rafts in *Abcc1*[−/−] mice. **a** Peritoneal macrophages from wild-type (WT) and *Abcc1*[−/−] mice were stained with anti-CD1d, Cholera toxin B (CTX) along with Hoechst for staining nucleus. Images were acquired using confocal microscopy. Figure shows representative images for the localization of CD1d in CTX[+] compartment in peritoneal macrophages at steady state. **b** Figure represents the number of Cholera toxin (CTX[+]) vesicles, average area of CTX[+] vesicles, and integrated density of CTX expression. **c** A compilation of the number of CD1d vesicles, average area of CD1d vesicles, and integrated density of CD1d expression on the cell surface. Representative images from one of two experiments with at least 100 cells per condition. Scale bar 10 μm. Graphs represent mean ± SEM. **p < 0.01; ***p < 0.001 (one-way ANOVA). See also Supplementary Fig. 13

iNKT cells. Although *Abca7* deficiency did not greatly reduce the immune response using iNKT hybridoma cells, our earlier work showed that *Abca7*[−/−] mice had a decreased lipid antigen response in vivo and macrophages from these mice had a reduced ability to present antigen to expanded lines of primary iNKT cells[57]. Therefore, the hybridoma cells may have provided a less sensitive readout than primary iNKT cells.

While Mrp1 had been reported to be a plasma membrane transporter mediating the efflux of anticancer drugs[58,59], because of its selective effect on GalGalCer responses, *Abcc1* gene knockdown also may have influenced aspects of the antigen

presentation process occurring deep in the endolysosomal system. Using inhibitor compounds, Mrp1 previously had been reported to be involved in the transport of sulfatide, a GSL, across the membrane of human DC[60], suggesting Mrp1, perhaps with other ABC transporters, could be a TAP-equivalent for transport of lipid antigens. The most profound alteration due to the absence of Mrp1, however, was in the trafficking of CD1d, not in antigen localization. There was reduced CD1d co-localization in both late endosomes and in lipid rafts, and it has been shown that disruption of these structures led to diminished iNKT cell stimulation[61–63]. Interestingly, loading of lipid antigens into CD1d in low

pH endosomal compartments is correlated with the appearance of these lipid antigen-CD1d complexes on the cell surface in lipid rafts[64,65], suggesting a connection between the lysosomal site of antigen loading and appearance of the antigen on the cell membrane in complex with CD1d in detergent resistant microdomains. The effects of Mrp1 deficiency were not selective for CD1d, however, as we detected decreases in the number and size of membrane lipid rafts and Lamp1+ vesicles. Overall, these findings indicate a larger role for Mrp1 in several aspects of lysosomal membrane biogenesis, consistent with its increased influence on GalGalCer responses.

Comparing our results to those from a genome-wide screen for elements affecting MHC class II presentation of peptides by human cells did not reveal any genes in common with the CD1d-mediated presentation pathway[66]. Neither screen was comprehensive, perhaps because as noted, the knockdowns were not complete, and also, a small number of residual antigen-antigen-presenting molecule complexes may be sufficient to activate T cells in some contexts. Nevertheless, the lack of overlap between the CD1d and MHC class II pathway is surprising. iNKT cells apparently do not use a TCR co-receptor, and therefore we speculate that iNKT cell activation may require increased clustering of CD1d in membrane microdomains or lipid rafts. While MHC class II presentation also has been shown to require lipid rafts, this was observed only at low antigen concentrations[67–69].

In summary, a genome-wide screen has identified a set of genes that positively regulate CD1d presentation of lipid antigens, including ABC family transporters and others involved in the endolysosomal network, including genes that had not been associated previously with antigen presentation in any system. Considering the conservation between mice and humans for CD1d antigen presentation[6], the selected genes from this screen may be applicable to efforts to modulate the activity of human iNKT cells. Furthermore, they may be implicated in presentation to other T lymphocytes reactive with lipid antigens presented by group I CD1 molecules.

## Methods

**Cell culture and reagents**. J774-CD1d, a mouse *Cd1d1.1* transfectant of the J774A.1 (American Type Culture Collection) macrophage cell line[70] and iNKT cell hybridoma DN3A4-1.2[71,72] were cultured at 37 °C under 5% CO$_2$ in Dulbecco's modified Eagle's medium containing 10% heat-inactivated fetal bovine serum (FBS), 100 U/ml penicillin, 100 U/ml streptomycin, 2 mM L-glutamine, 1 mM sodium pyruvate, and 10 mM HEPES. Antibodies used against various proteins along with their dilutions and catalog number are as follows: PE Rat Anti-Mouse CD1d (1:100, 553846); Mouse Anti-Mouse H-2K[d] (1:200, 553566); Purified Rat Anti-Mouse IL-2 (1:250, 554424); Biotin Rat Anti-Mouse IL-2 (1:1000, 554426); Biotin Rat Anti-Mouse CD8a (1:250, 553029); Biotin Rat Anti-Mouse CD19 (1:250, 553784); FITC Rat Anti-Mouse CD107a (LAMP1) (1:100, 553793); BV786 Rat Anti-Mouse CD45 (1:200, 564225); Brilliant Violet 605™; BV605 Rat Anti-Mouse CD19 (1:200, 563148); BV711 Hamster Anti-Rat/Mouse CD49a (1:200, 564863); and PE Rat Anti-Mouse IL-17A (1:200, 559502) from BD Biosciences. Anti-mouse CD8α Antibody (1:200, 100743); IFNγ Monoclonal Antibody PerCP-Cy5.5 (1:200, 45-7311-82); αGalCer:CD1d Complex Monoclonal Antibody (L363) (1:150, 12-2019-82); TER-119 Monoclonal Antibody, Biotin (1:250, 13-5921-85); TCR β Monoclonal Antibody, APC-eFluor 780 (1:200, 47-5961-82); Biotin CD24 Monoclonal Antibody (1:250, 13-0247-82); and CD103 (Integrin alpha E) Monoclonal Antibody (2E7), FITC (1:200, 11-1031-85) from ebioscience. Rat anti-mouse I-A/I-E Antibody (1: 200, 107622); Alexa Fluor® 700 anti-mouse CD4 Antibody (1:200, 100430); PE/Cy7 anti-human/mouse/rat CD278 (ICOS) Antibody (1:200, 313520); and APC anti-mouse CD183 (CXCR3) Antibody (1:200, 126512) from Biolegend. Rabbit Anti-Mouse Rab5 antibody (1:200, ab18211); Rabbit monoclonal to Tsg101 (1: 1000, ab125011); Rabbit polyclonal to Dock2 (1:1000, ab74659); Rabbit monoclonal to Vps11 (1:1000, ab170869); and Rabbit monoclonal to Snap29 (1:1000, ab138500) from Abcam. Goat anti-Rabbit AF488 secondary antibody (1:1000, A-11034) and Goat anti-Rat AF555 secondary antibody (1:1000, A-21434) from Invitrogen. Anti-rabbit IgG horseradish peroxidase (HRP)-linked (1:2500, 7074) and Anti-mouse IgG HRP-linked (1:2500, 7076) from Cell Signaling Technology. CD1d tetramer was prepared in-house. αGalCer was obtained from Kyowa Hakko Kirin Co. and GalGalCer was obtained from the Besra Lab, University of Birmingham. αGalCer-Bodipy was obtained from Besra Lab, University of Birmingham and from the Howell lab, University of Connecticut. For antigen

presentation and antigen uptake assays, αGalCer, GalGalCer, and αGalCer-Bodipy are used at 500 ng/ml concentration unless indicated. LIVE/DEAD™ Fixable Yellow Dead Cell Stain Kit (Thermo Fisher) was used to exclude dead cells during cell acquisition.

**Animals**. Mice were bred and housed under specific pathogen-free conditions in the vivarium of the La Jolla Institute for Allergy and Immunology (La Jolla, CA). Wild-type (WT; FVB) parental strain and *Abcc1*$^{-/-}$ mice were purchased from Taconic. All procedures were approved by the La Jolla Institute for Allergy and Immunology Animal Care and Use Committee. Mice of both genders were used and they were 8–12 weeks old.

**siRNA optimization**. We performed siRNA titrations against different target genes in J774-CD1d cells transfected with siGENOME SMARTpool (GE Dharmacon) siRNA using Interferin-HTS (Polypus) transfection reagent. Total RNA was extracted using the RNeasy Mini kit (Qiagen) according to the manufacturer's instructions. cDNA was synthesized using iScript (Bio-Rad) and real-time PCR was performed using SYBRgreen (Roche) on a LightCycler instrument (Roche). The first set of experiments, using a range of *Cd1d* siRNA concentrations from 1 to 100 nM, identified a 25 nM concentration as optimal, yielding a reduction of approximately 80% in *Cd1d* mRNA (Supplementary Fig. 1A). At this concentration of siRNA, there was approximately a 70–80% decrease in CD1d surface protein expression (Supplementary Fig. 1B). A kinetic analysis showed knockdown was maximal at 36–48 h (data not shown). We also tested knockdown of several kinase genes (*Camk2a*, *Mapk1*, *Prkcd*, *Mapk14*, and *Ptk2b*) at 25 and 50 nM concentration of siRNA. We observed decreases of 60–80% in mRNA levels (Supplementary Fig. 1C) using 25 nM siRNA, assessed using real-time PCR. Although a greater knockdown could be achieved at 50 nM, there also was increased cell death; therefore we selected 25 nM. Following primers are used for real-time PCR-

*Cd1d1*—F-CTGTCTGCGGGCTGTGAAAT, R-TCCCCAGAATCTCACGA CATATT

*Camk2a*—F-CCCTGGAATGACAGCCTTTGA, R-CCGGGACCACAGGTT TTCA

*Mapk1*—F-ACCAACCTCTCGTACATCGGA, R-AGCAAACCATGCCGTA GGC

*Prkcd*—F-AACCGTCGTGGAGCCATTAAA, R-GGCGATAAACTCGTGGT TCTTG

*Mapk14*—F-ACCTAGCTGTGAACGAAGACT, R-GTAGCCACGTAGCCTG TCATC

*Ptk2b*—F-AGGTATGACCTTCAAATCCGCT, R-CCGGAGCTGTTGATAAA AGTACA

**Western blot**. J774-CD1d cells were transfected with 25 nM siGENOME SMARTpool (GE Dharmacon) siRNA using Interferin-HTS (Polypus) transfection reagent. Thirty-six hours post transfection, cells were directly lysed in RIPA buffer supplemented with a protease and phosphatase inhibitor cocktail (Thermo Fisher) on ice for 30 min. Protein concentrations were measured using BCA assay kit (Thermo Fisher). Protein lysates were analyzed by SDS–polyacrylamide gel electrophoresis (Bio-Rad) and membranes were blotted with primary antibodies (Abcam), followed by secondary antibody (Cell signaling). Blots were developed using luminol reagent (Santa Cruz). Blots were stripped using stripping buffer (Cell signaling) and reprobed for loading control (β-actin—Cell Signaling). Quantitation on bands was performed using Quantity One software.

**Workflow of screen**. We used the mouse siGENOME siRNA Library SMARTpool (GE Dharmacon). J774-CD1d cells (3.5 × 10$^3$ cells/well) were first transfected with individual target-specific siRNA pools using reverse transfection in 384-well plates. A transfection mixture consisting of siRNA, Optimem (Gibco) and Interferin-HTS was prepared using a liquid handler Hamilton STAR and cells were plated using Wellmate (Thermo Fisher). Approximately 40 h post transfection, cells were loaded with GalGalCer at a concentration of 500 ng/ml for 6 h, and later antigen was removed and DN3A4-1.2 iNKT cell hybridoma cells (3.5 × 10$^3$/well) were added. These cells express a TCR encoded by Vβ8.2 (*Trbv13-2*) paired with the invariant Vα14 (*Trav11-Traj18*) α chain. After 15 h, 5 µl supernatant was used for an AlphaLISA-based IL-2 measurement (Perkin Elmer) using Beckman Coulter Biomek FXP liquid handler, and CellTitre blue (Promega) was added to the remaining cells to determine cell viability for 2 h. IL-2 release is a surrogate marker for TCR engagement. Each 384-well plate consisted of 2 wells each of the siGLO control, which is used for assessing siRNA uptake, a Tox siRNA control, which causes cell death and can be used to assess transfection efficiency, and negative controls, which contain two non-targeting siRNA pools. As a positive control for knockdown, we used Cd1d and B2m siRNAs. Every plate also had wells containing hybridoma cells and J774-CD1d cells either only with transfection reagent (cells only) or a transfection reagent along with antigen but containing no siRNA (cells+GalGalCer), and background-only wells containing just media (no cells) (Supplementary Fig. 1D). The primary screen was performed in duplicate. Alphalisa (Perkin Elmer) was performed using the collected supernatant according to the manufacturer's instructions. Briefly, anti-mouse IL-2 acceptor beads and biotinylated anti-IL-2 antibody were incubated with supernatant for 60 min. Streptavidin donor beads

were added for 30 min in the dark before reading the plate using EnVision plate reader (Perkin Elmer). CellTitre blue assay was performed for testing viability using EnVision plate reader (Perkin Elmer).

**Data analysis**. Analysis of every plate was performed with help from the bioinformatics core at the La Jolla Institute for Allergy and Immunology. First, a quality metrics was calculated based on positive and negative controls in each plate. Positive controls include *Cd1d* and *B2m* and a negative control was mock transfection followed by GalGalCer antigen (cells+GalGalCer). Plates were considered to have been assayed successfully if they showed a 70% decrease in the response to antigen when *Cd1d* and *B2m* expression were knocked down compared to controls. To exclude knockdowns that were generally toxic to the cells, knockdowns that showed cell viability below 50% of the control were excluded. We chose a minimum viability threshold of 50% based on this value being exceeded by 98% of all *B2m* and *Cd1d* knockdowns in different plates. We considered 30% or less production of IL-2 a significant knockdown based on 85% of the *B2m* and 73% of the *Cd1d* knockdowns achieving that threshold (Supplementary Fig. 1E), while none of the 201 wells with non-targeting siRNAs or cells with only transfection reagent in the entire screen achieved such a reduction.

**RNA isolation and microarray**. A microarray analysis was performed for the expression of genes in J774-CD1d cells and sorted peritoneal macrophages (F4/80$^+$) obtained directly from BALB/c mice. RNA was isolated using RNeasy mini kit (Qiagen) according to the manufacturer's instructions. Microarray was performed at UC San Diego Microarray Core facility using Affymetrix Mouse Gene 1.0 ST Array that interrogates 28,853 well-annotated genes with 770,317 distinct probes. All microarray data sets were processed in R, using customized scripts and Bioconductor modules. Raw intensity data were normalized with the "rma" function, with default parameters. A cutoff of 40, a measure of signal strength, was used to exclude genes expressed at a very low level. Microarray data were deposited in the Gene Expression Omnibus database. The assigned accession number is GSE109717.

**GO analysis**. We have employed the Bingo 2.44 plug-in into Cytoscape 2.8.3 to classify genes into different GO classes. In Bingo 2.44 we have used GO Slim generic for the clustering of genes, as determined by hypergeometric statistical test employing the Benjamini and Hochberg false discovery rate correction[21]. GO Slims are cut-down versions of the GO containing a subset of the terms in the whole GO.

**Ingenuity Pathway Analysis**. A core analysis was performed on identified genes using IPA Ingenuity Knowledge Base (Genes only, Version 01-13) reference set and all default settings. Networks of these genes were algorithmically generated based on their interrelationships. Experimentally observed direct relationships were used to generate these networks.

**Confocal microscopy and image processing**. Resident mouse peritoneal macrophages from WT FVB strain or *Abcc1*$^{-/-}$ mice were obtained by lavage of peritoneum with phosphate-buffered saline (PBS). After washing with PBS, cells were seeded on to a tissue culture plate in RPMI + 20% FBS for 2 h at 37 °C, supernatant was removed and adherent cells were seeded onto glass coverslips pre-coated with fibronectin (neuVitro) in 12-well tissue culture plates. Cells were fixed in 4% paraformaldehyde (Sigma) in PBS for 15 min at room temperature. Fixed cells were permeabilized using 0.2% Triton X-100 in PBS for 15 min, cells were then incubated in blocking buffer (10% FBS in PBS) for 1 h prior to a 45 min to 1 h incubation with respective primary antibodies (diluted using blocking buffer) followed by three washes with PBS/T (0.5%Tween 20 in PBS) and a single wash with PBS. Cells were further incubated with secondary antibody for 45 min, washed with PBS/T, and incubated with NucBlue® (Hoechst) Fixed Cell Reagent (Thermo Fisher) in PBS for 5 min for nuclear staining followed by three washes with PBS. For lipid raft staining we used Cholera Toxin Subunit B AF488 (Invitrogen) at a dilution of 1:1000 in PBS for 45 min before staining with Hoechst. All coverslips were mounted on slides with antifade (Thermo Fisher). A similar procedure was also performed for staining J774-CD1d cells after siRNA transfection. For lysosome staining, LysoTracker Red DND-99 (Thermo Fisher) was added at a dilution of 1:10 000 in cell culture media for the last 2 h of chase.

Multi-labeled slides of samples were imaged with an Olympus FV10i Laser Scanning Confocal microscope (Olympus, Center Valley, PA) or a Zeiss LSM 780 Laser Scanning Confocal Microscope (Zeiss, Thornwood, NY). Using either the FV10i or Zeiss Zen acquisition software, each circular coverslip containing cells was separated into three or four fields of view, which were acquired and a multi-tiled large image comprised of 3 × 3 or 4 × 4 panels were acquired per field. Each z series panel (1024 × 1024) was serially acquired with a ×60 or ×63 objective using a mechanical step size of 0.3 mm between sections and then stitched together through a 10% overlap with the Olympus FluoView 1000 (Olympus) or Zeiss Zen imaging software. Multi-labeled images were maximum projected and imported into Image Pro Premier (IPP) (Media Cybernetics, Inc MD) for further quantitative analysis, including co-localization assessment. For comparison, images were also imported into the Zeiss Zen software and analyzed using the Zen co-localization module. Quantitative analysis of staining intensity/dynamic range was obtained

after thresholding whereby we defined true signals based on control samples. These thresholds of dynamic range were used to obtain Manders correlation coefficients of the various paired stainings using the IPP and ZEN co-localization modules. Thresholds above background were also used in IPP to outline all cells (based on nuclear signal), which was dilated before the software to outline the entire cell. These whole cell outlines were defined as regions of interest from which all intracellular quantitative measurements of CD1d, Lamp1, and Rab5 signal were extracted, including average area, intensity, and integrated density. The parent-child module in IPP was used to define the cell outline as the parent and the children being the various intracellular markers of the trafficking pathway. Each of these fluorescently labeled organelles per cell were automatically outlined by the software and measurements extracted were imported into excel for further processing. An average of two or three experiments and 100–200 cells/condition are represented in figures.

**Mouse cell preparation**. Splenocytes and thymocytes were harvested by mechanical disruption on 70 μm nylon mesh followed by red blood cell (RBC) lysis and washing with RPMI supplemented with 10% FBS. For lung, tissue was digested with Stemcell spleen dissociation medium, and mechanically dissociated using GentleMacs Dissociator (Miltenyi). Tissues were strained though a 70 μm filter and washed with RPMI supplemented with 10% FBS followed by RBC lysis. For liver preparation, cells were harvested by mechanical disruption on 70 μm nylon mesh followed by 34% percoll gradient before RBC lysis and washing.

**Flow cytometry**. For staining of cell surface molecules, cells were suspended in staining buffer (PBS, 1% bovine serum albumin (BSA), and 0.01% NaN₃) and stained with fluorochrome-conjugated antibody at 0.1–1 μg/10$^7$ cells for 15 min in a total volume of 50 μl. FcγR-blocking Ab anti-CD16/32 (2.4G2, prepared in-house) was added to prevent nonspecific binding. Cells were analyzed with LSR II (BD Biosciences), and data were processed with DIVA (BD Bioscience) or Flow Jo (Tree Star, Ashland, OR) software.

**Mouse Vα14i NKT cell line**. Thymocytes from WT mice were enriched for Vα14i NKT cells by magnetic depletion using biotinylated antibodies against CD8α, CD19, CD24, and TER-119 (BD Biosciences and eBioscience) together with EasySep magnets and protocols and reagents from StemCell Technologies. Cells were then stained with αGalCer-loaded CD1d tetramers, together with anti-TCRβ antibodies, in staining buffer containing 1 μg/ml streptavidin. Tetramer-positive, TCRβ$^+$ cells were isolated using a FACSAria cell sorter (BD Biosciences). Sorted Vα14i NKT thymocytes were then cultured for 48 h at 10$^6$/ml in complete RPMI media on a plate coated with anti-TCRβ antibody together with soluble anti-CD28. Cells were then maintained by culture in complete RPMI media with 10 ng/ml mouse IL-15/IL15Ra (eBioscience) for 5 days before being used in experiments.

**Antigen presentation assay using Vα14i NKT cells**. Resident peritoneal macrophages were isolated from WT and *Abcc1*$^{-/-}$ mice. Cells were washed and plated onto a 96-well plate. Macrophages were then incubated with αGalCer and Gal-GalCer at concentrations indicated for 6 h. Antigen was washed away, and macrophages were incubated with the mouse Vα14i NKT cell line overnight. After 16 h, cell supernatants were collected and IFNγ was measured by ELISA (eBioscience) according to the manufacturer's protocol.

**Lipid injection in mice**. Mice were injected with 1 μg of αGalCer intravenously, and the sera of immunized mice were subjected to sandwich ELISAs to measure mouse IFNγ and IL-4 levels.

**S. pneumoniae infection**. *S. pneumoniae* URF918 were cultured in Todd-Hewitt broth (BD) at 37 °C in an incubator at 5% CO₂, collected at a mid-log phase and washed twice in PBS. For induction of pulmonary infection, mice were anesthetized with isoflurane and restrained on a small board. Mice were inoculated with *S. pneumoniae* ($3 \times 10^6$ to $1 \times 10^7$ colony-forming units in a volume of 50 μl/mouse) by insertion of a 24-gauge catheter into the trachea. For calculating the lung bacterial burden, tissues were collected at day 2 after infection and were homogenized in PBS. Homogenates were inoculated at different dilutions in a volume of 100 μl on 5% (vol/vol) sheep blood Mueller-Hinton agar plates and cultured for 18 h, followed by colony counting. For IFNγ staining after infection, lung cell suspensions were incubated for two hours at 37 °C in RPMI supplemented with 10% FBS and BD GolgiPlug (1:1000 dilution). Cells were washed and filtered again with RPMI + 10% FBS. Cells were resuspended in PBS supplemented with 2% BSA and stained for surface antibodies for 20 min on ice. Cells were permeabilized with ebioscience 10× perm buffer, diluted to 1× in dH2O, for 5 min. Cells were stained with intracellular antibodies overnight at 4 °C. Cells were washed and immediately run on flow cytometer.

**CD4$^+$ T-cell culture from DO11.10 TCR transgenic *Rag*$^{-/-}$ mice**. CD4+ cells were negatively selected using a combination of six biotinylated mAbs (anti-CD8α, CD11b, CD11c, NK1.1, B220, and TER-119) and anti-biotin microbeads (Miltenyi Biotec). Before the culture, 96-well plates were coated with 1.0 μg/ml anti-CD3

mAb diluted in PBS for 3–4 h at 37 °C. Purified cells were plated along with 1.0 µg/ml anti-CD28. After 3 days culture cells were re-stimulated with CD3 and IL-2 was added (20 ng/ml). Cells were IL-2-starved overnight before incubation with J774-CD1d cells.

**MHC-II antigen presentation assay**. J774-CD1d (10 000 cells/well) were transfected with various siRNAs in a U-bottom 96-well plate. Twenty-four hours later, cells were incubated with chicken OVA (Sigma) at a concentration of 1 mg/ml and expanded CD4+ T cells from DO11.10 TCR transgenic × $Rag^{-/-}$ mouse (20 000 cells/well). Forty-eight hours later IL-2 was measured in the supernatant by ELISA (BD Pharmingen).

## Data availability

Microarray data are deposited in Gene Expression Omnibus and the accession number is GSE109717. The authors declare that the main data supporting the findings of this study are within the article and its Supplementary Information files. Extra data are obtained from the corresponding authors upon request.

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

## Acknowledgements

We want to thank Zbigniew Mikulski for his help with microscopy and Jason Greenbaum for help with bioinformatics analysis. We would also like to thank Hilde Cheroutre for reviewing the manuscript and Gisen Kim and Hitosh Iwaya for advice on T-cell stimulation and expansion. This project has been funded in part with NIH grants AI45053 (M.K.), AI71922 (M.K.), RC4 AI092763-01 (M.K.), U01 GM111849 (A.R.H.), AI45889 (S.A.P.), and equipment grant S10RR027366 to the La Jolla Institute flow cytometry core facility. K.H. is supported by Research fellowships of Japan Society for the Promotion of Science for young scientists' and postdoctoral fellowship from the Uehara memorial foundation. C.M.C. is supported by American Lung Association Senior Research Training Fellowship RT-412662. M.Z. is supported by NIH training grant T32 AR064194.

## Author contributions

S.C., J.G., S.S., and M.K. designed the experiments, which were performed by S.C., J.G., W.B.K., A.K., K.H., C.M.C., and M.Z. Data analysis was performed by S.C., A.C., Z.F., and B.P. N.V., S.K.R., S.A.P., G.B., and A.R.H. provided reagent. S.C. and M.K wrote the manuscript. All authors reviewed the manuscript.

## Additional information

**Competing interests:** The authors declare no competing interests.

