## [Peer Review File · Nature Communications]

Reviewers' comments:

Reviewer #1 (Remarks to the Author):

The manuscript by Chandra et al. describes the results of a whole genome siRNA screen to elucidate the CD1d bio-synthetic pathway controlling invariant NKT cell activation. The results are of importance, as the mechanisms controlling trafficking of CD1d molecules to the cell surface and their loading with lipid antigens remain unclear. The paper is strengthened by the use of the L363 antibody, to detect the effect of the knockdowns on surface expressed CD1d/lipid complexes, and of α GC-Bodipy, to look at lipid trafficking. However, several aspects of the manuscript need to be strengthened and additional experiments are required.

Major Issues:

Figure 2 and Supplementary Fig. 2B. Since several knockdowns have an effect on lipid rafts and LAMP1 expression, specificity controls are needed to demonstrate whether or not the effect is CD1d specific (line 379). Results shown in Supplementary Figure 2B, only demonstrating the level of expression of MHC-II expression, are not sufficient, as they only indicate the level of cell surface expression of MHC-II molecules, rather than presentation of MHC-II loaded peptides.

Figure 3A. Western blot of individual genes, with loading controls, should be shown. It would be helpful to show representative FACS plots indicating CD1d and CD1d/ α GC expression on the cell surface.

Figure 4A and B. Western blot of individual genes, with loading controls, should be shown. In Figure 4B, why is the Tsg101 condition not significantly (ns) different from the Control? The error bar is so small and the mean is so different.

Figure 4C: LAMP-1 staining is very variable and co-localization comparisons to LAMP-1 are made when the LAMP-1 is also been affected by the treatment. Although the authors mention that this could be a consequence of the Tsg101 reduced expression treatment (lines 241-242), in the images they have shown, the LAMP-1 staining looks different in the Snap29, Vps11 and Dock2 conditions also. It would be appropriate to quantify the distribution of LAMP-1 for each condition as in Figure 6.

Figure 5B. Western blot of Abcc1 with loading controls should be shown. Histograms to assess whether down-regulation of CD1d-lipid complexes is unimodal or bimodal should be shown.

Figure 5C.

- Comparison of the level of CD1d expression in peritoneal macrophages from Abcc1 $-/-$ and WT mice should be shown.
- Have peritoneal macrophages been loaded with GGC (as indicated in the figure legend) or with α GC (as indicated in the figure)? Based on the results of Figure 6 indicating that alterations in late endosomes and lysosomes are likely to contribute to the pronounced effect of Abcc1 knockdown on the response to GGC (297), Figure 5C should show results obtained with GGC and as control with α GC.

Figure 5D. Results obtained with a range of α GC and GGC concentrations should be shown.

Figure 6: The authors need to say how they performed this analysis. It is not written in the methods.

Figure 7: The authors need to say how they performed this analysis. It is not written in the methods.

Correlation coefficients. The authors have used the Mander's test (line 915), which works best for sparse regions of fluorescence, but not for fluorescence distributions, which span the whole cell. The Pearson's test would be more appropriate, as it takes into account every pixel, not only those above a threshold value. The only issue with the Pearson's test is that non-cell pixels need to be excluded, but this could be achieved quite easily, by thresholding the DAPI signal, which is independent of the gene knock-down.

Minor Issues:

Figure 3. Y axis needs to be explained. Absolute values in control samples should be indicated in the legend.

Legend of Figure 3. Where are the p values shown?

Several grammatical problems/inconsistencies in the spelling of words (e.g. labeled and labelled); or (905): "...each circular coverslip of cells was separated into three or four number of fields of view were acquired".

Reviewer #2 (Remarks to the Author):

Chandra et al. Nature 2018 submission

In their manuscript, "Novel components involved in lipid antigen presentation: a role for the multi-drug resistance proteins," the Chandra et al. employ a genome-wide siRNA screen in a murine macrophage cell line to identify novel factors involved in CD1d lipid antigen presentation. After validating a subset of genes identified in their screen for their ability to modulate CD1d lipid antigen presentation through a mechanism that involves trafficking of CD1d to lysosomes, the authors focus their study on one target, Mrp1 encoded by Abcc1. The authors ascribe a novel function of Mrp1, a component of the multidrug resistance complex in regulating the activation of CD1d-restricted iNKT cells after infection with *Streptococcus pneumoniae* and demonstrate that deficiencies in Mrp1 lead to increased sensitivity to bacteria infection compared to wildtype hosts, consistent with a role for this protein in regulating iNKT dependent inflammatory responses. This is an extremely well done study of significant importance to the field.

Major comments

The authors do a thorough job performing their genome-wide screen and secondary validation of targets that significantly disrupt CD1d-restricted responses using assays to measure CD1d expression, trafficking and antigen presentation. While the authors do some gene ontology analyses to identify putative pathways associated with the 48 targets they validate, it is unclear how these targets are functionally related. For instance 4 targets they evaluate in detail (Dock2, Snap29, Tsg101, Vps11) appear to target CD1d responses in distinct ways that may or may not be mechanistically related. It would be helpful if the authors could draw some inferences or make some conclusions about a global mechanistic view of how these various targets may interact in a systematic manner.

The authors hypothesize that deficiencies in Abcc1, encoding Mrp1, provide protection against infection by *S. pneumoniae* by limiting iNKT inflammatory responses based on their observation that knockdown of Abcc1 leads to reduced formation of CD1d-lipid complexes at the cell surface of macrophages. In a separate study, it has been shown that MRP1 deficient mice are protected against *S. pneumoniae* infection through a mechanism involving decreased efflux of leukotrienes, such as LTC₄, more consistent with its classical transporter function (Schulz et al. 2001). On the

other hand, other studies have purported that MRP1 deficient animals are more sensitive to *M. tb* infection due to reduced Th1 response (Verbon et al. 2002). *M. Tb* also possesses putative CD1d-restricted antigens (Sada-Ovalle et al. 2008, Chackerian et al. 2002). Although the responses that they observe are consistent with their hypothesis, it would be appropriate to entertain alternative hypotheses for the nature of the protection observed and discuss these possibilities in the manuscript.

Minor Comments

Figure 5F is missing from the manuscript

RESPONSE TO THE REVIEWERS

We thank the reviewers for their time and effort, which has helped us to improve the manuscript. Below please find our responses to the reviewers' comments. We have provided a revised version of the manuscript that takes into account the reviewers' comments and changes to the manuscript text are indicated with yellow highlighting for ease of re-review. We thank you for consideration of our work and hope that our revised manuscript is now suitable for publication in *Nature Communications*.

Reviewer #1

Major Issues:

Question: Figure 2 and Supplementary Fig. 2B. Since several knockdowns have an effect on lipid rafts and LAMP1 expression, specificity controls are needed to demonstrate whether or not the effect is CD1d specific (line 379).

Response: There can be cell-to-cell and image-to-image variability, but in new Supplementary Figure 8, we analyzed > 200 cells in each case and quantified the effect of four gene knockdowns (*Dock2*, *Tsg101*, *Vps11*, *Snap29*) on the total area and number of Lamp1⁺ vesicles in J774-CD1d cells. For these four gene knockdowns, there was no change in area, but a statistically significant increase in the number of vesicles only when *Dock* was knocked down. This was not the case for *Abcc1* knockdown, which affected the number and size of vesicles. Regardless, we are not implying that effects are totally CD1d specific. Other processes could be affected and MHC class II presentation has now been analyzed in detail.

Question: Results shown in Supplementary Figure 2B, only demonstrating the level of expression of MHCII expression, are not sufficient, as they only indicate the level of cell surface expression of MHCII molecules, rather than presentation of MHCII loaded peptides.

Response: We have performed MHC class II-restricted antigen presentation assays using OVA antigen, expanded CD4⁺ T cells from TCR transgenic DO11.10 x *Rag*^{-/-} mice, and J774-CD1d antigen presenting cells with each of the 48 genes separately knocked down. These data are now included as new Supplementary Figure 5. Considering the gene knock downs that decreased MHC class II presentation, we found that there is only limited overlap with the lipid antigen presentation pathway we have analyzed.

Question: Figure 3A. Western blot of individual genes, with loading controls, should be shown. It would be helpful to show representative FACS plots indicating CD1d and CD1d/ α GC expression on the cell surface.

Response: We have now included western blots of the four individual proteins that we focused on along with non-targeting siRNA and transfection reagent controls. The data are shown in new Supplementary Figure 7 and show the efficacy of the knockdowns. Flow cytometry plots showing surface CD1d and CD1d/ α GC are shown in new Supplementary Figure 6.

Question: Figure 4A and B. Western blot of individual genes, with loading controls,

should be shown. In Figure 4B, why is the Tsg101 condition not significantly (ns) different from the Control? The error bar is so small and the mean is so different.

Response: We have now included western blot as described above. We have now included a new experiment for Lamp1-CD1d co-localization following *Tsg101* knockdown, but in this experiment as well the p value does not reach significance ($p=0.07$) (Figure 6B). We have also includes new images taken from LSM 780 in Figure 6C.

Question: Figure 4C: LAMP1 staining is very variable and colocalization comparisons to LAMP1 are made when the LAMP1 is also been affected by the treatment. Although the authors mention that this could be a consequence of the Tsg101 reduced expression treatment (lines 241,242), in the images they have shown, the LAMP1 staining looks different in the Snap29, Vps11 and Dock2 conditions also. It would be appropriate to quantify the distribution of LAMP1 for each condition as in Figure 6.

Response: We have now quantitated the average area and size of Lamp1⁺ vesicles in various gene knockdowns and it is shown in Supplementary Figure 8. We have not observed a significant change except that there are more Lamp1⁺ vesicles in *Dock2* knockdown cells.

Question: Figure 5B. Western blot of Abcc1 with loading controls should be shown.

Response: We have tested several antibodies for *Abcc1* by western blot but couldn't detect the protein. However, analysis of cells from the *Abcc1* knockout mice validate our conclusion regarding CD1d-mediated antigen presentation.

Question: Histograms to assess whether downregulation of CD1d lipid complexes is unimodal or bimodal should be shown.

Response: Flow cytometry histograms are now included in new Figure 8C. The expression is essentially unimodal.

Question: Figure 5C.

Comparison of the level of CD1d expression in peritoneal macrophages from *Abcc1* / and WT mice should be shown.

Response: We now have included these data in new Supplementary Figure 10A. There is no difference in surface CD1d expression by F4/80+ peritoneal macrophages.

Question: Have peritoneal macrophages been loaded with GGC (as indicated in the figure legend) or with α GC (as indicated in the figure)? Based on the results of Figure 6 indicating that alterations in late endosomes and lysosomes are likely to contribute to the pronounced effect of *Abcc1* knockdown on the response to GGC (297), Figure 5C should show results obtained with GGC and as control with α GC.

Response: Thank you for pointing that out. This inconsistency has been corrected in the figure legends. We now are showing a side-by-side comparison of the effect of Abcc1 deficiency on IFN γ secretion upon stimulation with α GC or GGC at various concentrations in new Fig. 8D. Although in the screen, the GGC response was more affected by Abcc1 knockdown than the α GC response, this is not so evident in the figure. The experimental systems are very different, however, with gene knock down in a transformed macrophage line and hybridoma readout in one cases versus primary macrophages, gene deficiency and primary cell lines in the other. We note that in some experimental contexts presentation of α GC is also highly dependent on internalization into APCs, although this is not always the case.

Question: Figure 5D. Results obtained with a range of α GC and GGC concentrations should be shown.

Response: This is a good suggestion, but we cannot obtain enough mice to perform this experiment. We used to buy these mice from Taconic, and according to their terms and conditions, we were not allowed to breed them. Taconic has now discontinued this strain, and they only are available from EZcohort who would need to derive the mice from frozen stock. Not only would this delay the paper significantly, but the price they re requesting for this service (\$15k) is exorbitant. Therefore, we are unable to perform this experiment.

Question: Figure 6: The authors need to say how they performed this analysis. It is not written in the methods.

Response: We have now included a description of how the confocal microscopy data were analyzed (Lines- 1029 to 1038).

Question: Figure 7: The authors need to say how they performed this analysis. It is not written in the methods.

Response: We have now included a description of how the confocal microscopy data were analyzed (Lines- 1029 to 1038).

Question: Correlation coefficients. The authors have used the Mander's test (line 915), which works best for sparse regions of fluorescence, but not for fluorescence distributions, which span the whole cell. The Pearson's test would be more appropriate, as it takes into account every pixel, not only those above a threshold value. The only issue with the Pearson's test is that non cell pixels need to be excluded, but this could be achieved quite easily, by thresholding the DAPI signal, which is independent of the gene knockdown.

Response:

Mander's coefficient allows us to separate the direction of analysis that the two signals are occupying in the same space: that is, how much CD1d signal goes to the Lamp1/Rab5 compartment (M1) and how much Lamp1 is occupied by CD1d (M2), of which we utilize the former taking into account the distribution pattern of CD1d throughout the cell. This makes Manders more applicable to how we are defining colocalization, irrespective of the levels of expression of the two variables being compared. In Pearson's, the analysis

is irrespective of the direction. Pearson's coefficient is only reliable for high correlations and therefore Pearson would be ideal if we were looking for potential binding partners or proteins that cluster close enough to FRET. These considerations make Mander's more applicable as to how we are defining co-localization.

References:

(1) Kenneth W. Dunn, 1 Malgorzata M. Kamocka,1 and John H. McDonald2:

A practical guide to evaluating colocalization in biological microscopy *Am J Physiol Cell Physiol.* 2011 Apr; 300(4): C723–C742. Published online 2011 Jan 5. doi: 10.1152/ajpcell.00462.2010

(2) John H. McDonald* and Kenneth W. Dunn†: Statistical tests for measures of colocalization in biological microscopy; *Microsc.* 2013 December ; 252(3): 295–302. doi:10.1111/jmi.12093.

Question: Minor Issues:

Figure 3. Y axis needs to be explained. Absolute values in control samples should be indicated in the legend.

Legend of Figure 3. Where are the p values shown?

Response: Thank you for pointing this out. The y axis is labeled now and we have included the p-values in the figure.

Question: Several grammatical problems/inconsistencies in the spelling of words (e.g. labeled and labelled); or (905): "...each circular coverslip of cells was separated into three

or four number of fields of view were acquired”.

Response: Thank you for this comment. We have improved the writing of the manuscript.

Reviewer #2

Question: The authors do a thorough job performing their genomewide screen and secondary validation of targets that significantly disrupt CD1drestricted responses using assays to measure CD1d expression, trafficking and antigen presentation. While the authors do some gene ontology analyses to identify putative pathways associated with the 48 targets they validate, it is unclear how these targets are functionally related. For instance 4 targets they evaluate in detail (Dock2, Snap29, Tsg101, Vps11) appear to target CD1d responses in distinct ways that may or may not be mechanistically related. It would be helpful if the authors could draw some inferences or make some conclusions about a global mechanistic view of how these various targets may interact in a systematic manner.

Response: Thanks for your comments. We have now included pathway analysis using IPA in new Figure 3 and new Supplementary Figure 2. We could cluster the 48 genes into three functional networks. For example, one of the three, which encompassed half of the hits, is related to protein trafficking and cellular organization and it includes members of the HOPS and SNARE complexes that interact directly. The analysis of all three sub-networks is discussed in the revised manuscript.

Question: The authors hypothesize that deficiencies in *Abcc1*, encoding *Mrp1*, provide protection against infection by *S. pneumonia* by limiting iNKT inflammatory responses based on their observation that knockdown of *Abcc1* leads to reduced formation of CD1d lipid complexes at the cell surface of macrophages. In a separate study, it has been shown that MRP1 deficient mice are protected against *S. pneumonia* infection through a mechanism involving decreased efflux of leukotrienes, such as LTC₄, more consistent with its classical transporter function (Schulz et al. 2001). On the other hand, other studies have purported that MRP1 deficient animals are more sensitive to *M. tb* infection due to reduced Th1 response (Verbon et al. 2002). *M. Tb* also possesses putative CD1d-restricted antigens (SadaOvalle et al. 2008, Chackerian et al. 2002). Although the responses that they observe are consistent with their hypothesis, it would be appropriate to entertain alternative hypotheses for the nature of the protection observed and discuss these possibilities in the manuscript.

Response: Thank you for pointing this out. In Schulz et al. paper, the background of their mice is (129/Ola)/FVB (50%/50%) whereas the mice that we used in our study or in study by Verbon et al. were FVB background. Also, the infection route they used was intranasal, and they used a 50-250-fold lower dose (10^4), whereas we were using a retropharyngeal route. Furthermore, their *S. pneumoniae* serotype was different. Any one of the above or a combination of the differences might have lead to the discrepant results. We have now mentioned this in discussion.

Question: Minor Comments-

Figure 5F is missing from the manuscript

Response: Thanks for pointing this out. Somehow the letter went below the plot, but we have corrected it now.

REVIEWERS' COMMENTS:

Reviewer #1 (Remarks to the Author):

The authors have satisfactorily addressed all my concerns and the revised manuscript has been significantly strengthened.